# Towards LLM4Floorplan: Agents Can Do What Engineers Do in Chip Design

## Abstract

Open-source tools have actively propelled advancements in physical electronic design, yet the deployment still requires substantial expertise. Recent progress in large language model (LLM)-based agents offer potential for automating physical design, but challenges remain in imparting domain-specific expertise and extracting case-specific design objectives to meet complex requirements. To address these issues, we introduce LLM4Floorplan, a multi-agent **Floorplan**ner powered by **LLM**s. Unlike flow-level approaches that design workflows for multiple tasks, LLM4Floorplan is the first task-level agent specifically dedicated to a single physical design task. Specifically, we propose a simple yet effective search-cluster-based retriever that extracts the most relevant and diverse solutions from prior knowledge, drawing on essential domain-specific knowledge to ensure robust design performance. Building on the retriever, LLM4Floorplan integrates a novel Dynamic Retrieval-Augmented Thought (DRAT) prompting technique in which the LLM generation interacts with the retrieval system to precisely capture case-specific design objectives. With these innovations, LLM4Floorplan simulates the workflow of human engineers by facilitating task comprehension, model selection, hyperparameter tuning, code revisions, and performance evaluation. Extensive evaluations on public circuits with seven different LLM backbones demonstrate that LLM4Floorplan exhibits strong task comprehension and decision-making capabilities. Remarkably, for the strict requirement, LLM4Floorplan boosts the success rate from 0.250 to 0.875.

## 1 Introduction

Physical design, including floorplanning (Knechtel et al., 2015; Li et al., 2022), placement (Chiou et al., 2016; Liao et al., 2023; Cheng et al., 2022), routing (Liu et al., 2013; Du et al., 2023), etc., in electronic design automation (EDA) is critical in the design of very large-scale integration (VLSI) and attracts a lot of effort from classical (Knechtel et al., 2015; Chiou et al., 2016; Liu et al., 2013) and machine learning solutions (Li et al., 2022; Liao et al., 2023; Cheng et al., 2022; Du et al., 2023). However, it still highly depends on the expertise and consuming time of human engineers to perform a chain of tasks as shown in Fig. 1, which significantly sacrifices the automation and efficiency in the industry.

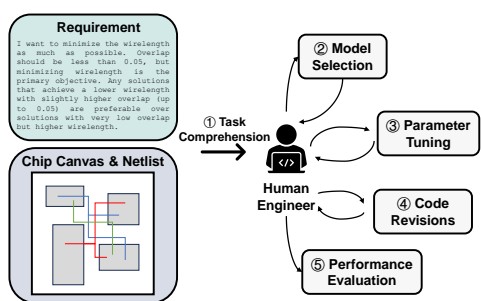

Figure 1: Example of the design workflow of floorplanning. It currently relies heavily on experienced human engineers for task comprehension, model selection, hyperparameter tuning, code revisions, and performance evaluation.

Recently, with the rapid development of large language models (LLMs) (Achiam et al., 2023; Touvron et al., 2023; Anthropic, 2024), especially the emergence of LLM-powered agent systems in various scenarios like video game (Tan et al., 2024), smartphone users (Yang et al., 2023b), software development (Qian et al., 2024), the potential of EDA agents has garnered significant attention from both industry and academia (Wang et al., 2024c). However, in contrast to other agent systems, EDA agents face two critical **challenges**: 1) imparting sufficient domain-specific expertise to handle

the intricacies of physical design, and 2) extracting case-specific design objectives to meet diverse and complex requirements. Domain-specific expertise provides the foundation for understanding and tackling general EDA challenges (e.g., floorplanning principles), while case-specific objectives focus on how this knowledge is applied to meet the unique needs of an individual design.

To address these, we target floorplan, a critical stage in physical design, and devise LLM4Floorplan, a multi-agent floorplanner powered by LLMs, to perform domain- and case-specific guidance, which designs floorplan layouts that successfully meet diverse requirements.

Specifically, inspired by RAT (Wang et al., 2024d), we develop a retrieval system (retriever) to extract domain-specific prior knowledge. Unlike RAT, which retrieves information via web searches, this approach is impractical for EDA design due to the closed-source nature of the community. To address this, we construct a database that stores historical design experience and apply a simple search and clustering technique to obtain relevant, diverse instances as domain-specific prior knowledge. Building on the retriever, we propose a novel Dynamic Retrieval-Augmented Thought (DRAT) prompting technique, enabling LLMs to interact with the retriever. With this technique, LLM4Floorplan captures case-specific design objectives, facilitating the design of floorplan layouts under complex requirements. These innovations allow LLM4Floorplan to reduce the labor-intensive efforts of human engineers across various design tasks, including task comprehension, model selection, hyperparameter tuning, code revisions, and performance evaluation.

To assess the efficacy of these techniques, we propose a benchmark comprising six distinct floorplanning requirements. Empirical results on public datasets using seven LLM backbones demonstrate improved performance and successful design outcomes. Notably, LLM4Floorplan achieves successful designs even under complex and unseen requirements, showcasing its strength in generality and potential for more personalized design solutions. **The highlights of this work are:**

1) **Pioneering Task-Level Multi-Agent for Physical Design.** We introduce LLM4Floorplan, to the best of our knowledge, the first implementation of a multi-agent system dedicated to a specific physical design task. It builds on large language models (LLMs) to automate complex design processes, providing a novel framework for floorplanning and reducing manual effort in design processes.

2) **Novel Retrieval System to Integrate Domain-Specific Design Guidance.** We introduce a novel retrieval system featuring a search and clustering module that extracts the most relevant and diverse instances as domain-specific prior knowledge.

3) **DRAT Prompting for Case-Specific Design Guidance.** Building on the retrieval system, we propose a novel DRAT prompting technique, which enables LLM4Floorplan to dynamically integrate the retriever and LLM generation. DRAT enhances the system's ability to capture case-specific design objectives, addressing diverse and complex design challenges effectively.

4) **Benchmark and Significant Empirical Performance.** We validate our approach by introducing a new benchmark with six distinct design requirements and testing LLM4Floorplan on floorplanning tasks using public circuits from the MCNC and GSRC datasets with seven LLMs. Our results demonstrate that LLM4Floorplan consistently outperforms existing floorplanning baselines, particularly in strict design requirements, where the success rate improves from 0.250 to 0.875.

## 2 RELATED WORK

**LLM prompting.** Since the emergence of pretrained LLMs as foundation models, prompting engineering has been explored to improve answer quality, serving as an alternative to fine-tuning (Cobbe et al., 2021). Among various prompting techniques, in-text learning (Dong et al., 2022) enables LLMs to learn tasks from only a few examples provided as demonstrations. To enhance LLM capabilities in solving math word problems, Chain-of-Thought (CoT) (Wei et al., 2022) incorporates intermediate reasoning steps, while Retrieval-Augmented Generation (RAG) (Lewis et al., 2020) retrieves information from an external knowledge base. Combining the strengths of both CoT and RAG, Retrieval-Augmented Thoughts (RAT) (Wang et al., 2024d) addresses long-horizon generation and improves rating scores. However, RAT lacks an external knowledge base when applied directly as an EDA agent due to the closed-source nature of the EDA community. To overcome this limitation, we propose a novel dynamic RAT prompting technique, along with a search-clustering-based retriever, to extract both domain- and case-specific guidance.

**LLMs as Agents.** Employing LLMs as specialized agents (Wang et al., 2024b; Hu et al., 2024) is becoming more popular to address entertainments or industrial problems. For example, Cicero (FAIR et al.,2022), CRADLE (Tan et al., 2024), Park et al. (2023), and Xu et al. (2023) regard LLM as game characters or players to interact with gaming environments. ToolLLM (Qin et al., 2023), Toolformer (Schick et al., 2023), GPT4Tools (Yang et al., 2023a), and ToolkenGPT (Hao et al., 2023) instruct LLMs to use external tools. AppAgent (Yang et al., 2023b) and Mobile-Agent-v2 (Wang et al., 2024a) use LLM to simulate smartphones users. Additionally, Codex (Chen et al., 2021), App-World (Trivedi et al., 2024) Chatdev (Qian et al., 2024) use LLMs to develop software. These agents are generally powered by frequent interactions with environments and abundant datasets, which are intractable to be applied to EDA scenarios.

**LLM for EDA.** LLM for EDA (Zhong et al., 2023; Wang et al., 2024c) recently attracts lots of attention. Among these efforts, most of the LLM-based approaches are devised to generate Hardware Description Language (HDL) code such as Verilog (Blocklove et al., 2023; Liu et al., 2023b; Lu et al., 2024) or Register Transfer Level (RTL) code (Fu et al., 2023; Wan et al., 2024), which mainly equip LLMs' strong language capability. Additionally, ChipGPT (Chang et al., 2023), BetterV (Pei et al., 2024) and DeLorenzo et al. (2024) are proposed for Verilog design optimization while Analog-Coder (Lai et al., 2024) uses LLMs to generate codes to design analog circuits. The most related works to our paper are ChipNeMo (Liu et al., 2023a) and ChatEDA (Wu et al., 2024), which design agents to interact with EDA tools. However, these agents operate at the flow level, designing high-level workflows without significantly contributing to the enhancement of specific tasks, whereas our task-level agent aims to improve a specific task. Our LLM4Floorplan is a task-level agent that emphasizes floorplanning, which is a prior and critical task in physical design.

# 3 PRELIMINARIES AND PROBLEM FORMULATION

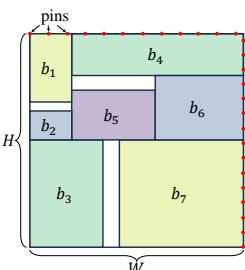

Figure 2: Example of a floorplanning design.

**Floorplan.** Floorplanning is a prior stage in physical design and recognized as an NP-hard Murata et al. (1996) problem, where the fixed-outline floorplanning formulation is current well-established formulation (Li et al., 2022). Specifically, as is shown in Fig. 2, the layout region is defined as a rectangular area with a given width $W$ and height $H$, extending from coordinates $(0, 0)$ in the lower-left corner to $(W, H)$ in the upper-right corner. Additionally, a netlist $(\mathcal{V}_b, \mathcal{V}_p, \mathcal{E})$ is provided, where each element $b_i \in \mathcal{V}_b, (i = 0, 1, \ldots, n_b - 1)$ represents a block (rectangle) with fixed width $w_i$ and height $h_i$, and area $a_i = w_i \cdot h_i$. The center of block $b_i$ is positioned at $(x_{b_i}, y_{b_i})$. Similarly, each $p_i \in \mathcal{V}_p, (i = 0, 1, \ldots, n_p - 1)$ corresponds to a pin (point) with fixed coordinates $(x_{p_i}, y_{p_i})$. Each net $e_i \in \mathcal{E}, (i = 0, 1, \ldots, n_e - 1)$ connects a subset of blocks and pins, denoted as $e_i = \{b_1^{(e_i)}, b_2^{(e_i)}, \ldots, p_1^{(e_i)}, p_2^{(e_i)}, \ldots\}$. For each net $e_i$, the Half-Perimeter Wire Length (HPWL) is calculated as:

$$\text{HPWL}(e_i) = \max_{b \in e_i}(x_b) - \min_{b \in e_i}(x_b) + \max_{b \in e_i}(y_b) - \min_{b \in e_i}(y_b). \tag{1}$$

The objective of the fixed-outline floorplanning problem is to optimize the locations and shapes of all movable blocks, minimizing total HPWL across all nets while ensuring no overlapping area among blocks, maintaining suitable aspect ratios, and keeping all blocks within the fixed outline. The optimization variables $\Theta = \{(w_i, h_i, x_{b_i}, y_{b_i})\}_{i=0}^{n_b-1}$ include width, height $(w_i, h_i)$ and coordinates $(x_{b_i}, y_{b_i})$ for each block $b_i \in \mathcal{V}_b$. Further details on floorplanning are provided in Appendix B.

**RAT prompting.** Retrieval-Augmented Thought (RAT) (Wang et al., 2024d) is a mitigation of Chain-of-Thoughts (CoT) (Wei et al., 2022) and Retrieval-Augmented Generation (RAG) (Lewis et al., 2020). Specifically, given a task prompt $\mathcal{P}$ and a powerful LLM $f_{\boldsymbol{\theta}}$ with pretrained parameters $\boldsymbol{\theta}$, CoT generates *zero-shot* thoughts $\mathcal{T} = \{\boldsymbol{t}_i\}_{i=1}^{T} \sim f_{\boldsymbol{\theta}}(\cdot|\mathcal{P})$ with $T$ thought steps, based on which RAT generates the thought steps $\tilde{\boldsymbol{t}}_{1:i}$ according to comprehensive conditional distributions:

$$\tilde{\boldsymbol{t}}_{1:1} = \tilde{\boldsymbol{t}}_1 \sim f_{\boldsymbol{\theta}}(\cdot|\mathcal{P}, \boldsymbol{t}_1, \boldsymbol{r}_1), \quad \tilde{\boldsymbol{t}}_{1:i} \sim f_{\boldsymbol{\theta}}(\cdot|\mathcal{P}, \tilde{\boldsymbol{t}}_{1:(i-1)}, \boldsymbol{t}_i, \boldsymbol{r}_i) \ (2 \leq i \leq T), \tag{2}$$

where $\boldsymbol{r}_i$ is relevant documents retrieved by the query $\boldsymbol{q}_i = g_{\boldsymbol{\phi}}(\mathcal{P}, \boldsymbol{t}_{1:i})$. The query function $g_{\boldsymbol{\phi}}$ with parameters $\boldsymbol{\phi}$ is a text encoder or LLM that translates the task prompt $\mathcal{P}$ and the thought steps $\boldsymbol{t}_{1:i}$ into a query $\boldsymbol{q}_i$, allowing the retrieval system to deal with it. With the thought steps and retrieval system, RAT significantly enhances the reasoning ability of LLMs.

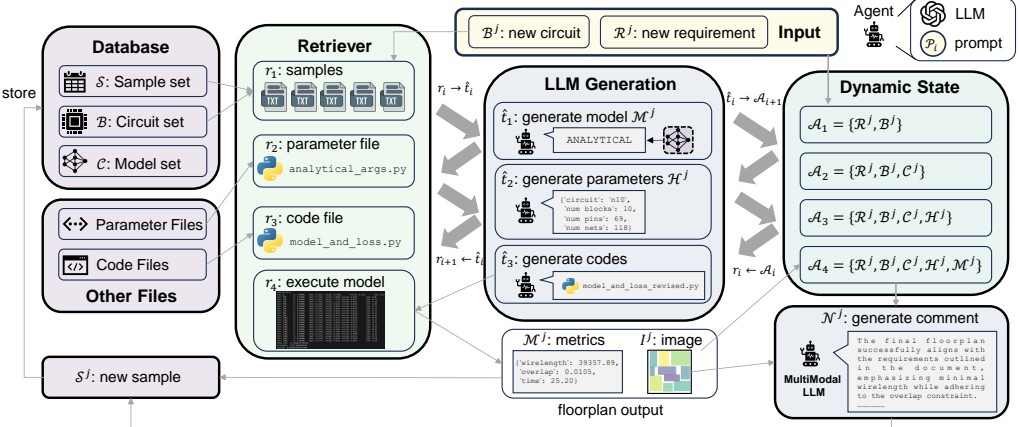

Figure 3: The pipeline of LLM4Floorplan consists of four main components: data, the retrieval system, LLM agents, and the dynamic state. Upon receiving input, the agents retrieve necessary materials from the database, related documents, or by executing the floorplan approach, while maintaining a dynamic state to track the design schedule. After the design is completed, the comment agent evaluates the performance and adds a new instance to the database.

## 4 LLM4FLOORPLAN

**Overview.** In this section, we propose LLM4Floorplan, a floorplanner powered by LLM, to address two challenges for EDA agents: 1) *impart domain-specific expertise*, and 2) *extract case-specific design objectives*. Specifically, we respectively introduce an effective retriever in Sec. 4.1 and a novel prompting technique in Sec. 4.2, and give a specific process of LLM4Floorplan in Sec. 4.3. We show the pipeline in Fig. 3, where LLM4Floorplan contains four aspects, including datasets, retrieval system, decisions, and dynamic information state process.

### 4.1 SEARCH-CLUSTER-BASED RETRIEVER

To address the first challenge, we initially construct a standard database $\mathcal{D} = (\mathcal{S}, \mathcal{B}, \mathcal{C})$ that contains the set of executed instances $\mathcal{S}$, the set of circuit information $\mathcal{B}$, and the set of model selections $\mathcal{C}$. Specifically, we design an instance collection system. Each time an experiment is conducted, the corresponding log is recorded as an instance $\mathcal{S}^j \in \mathcal{S}$. These records serve as an experience pool, allowing LLMs for retrieving domain-specific knowledge for improved decision.

The retrievers in RAG (Lewis et al., 2020) and RAT (Wang et al., 2024d) are quite important in the system. Unlike RAG, which fine-tunes a pretrained retriever and generator, LLM4Floorplan has limited instances from which to learn parametric knowledge. Therefore, to acquire better knowledge, we develop a search-cluster-based retriever that selects instances based on *relevance* and *diversity*.

Specifically, in the retriever, we employ a text encoder $g_\phi$ to obtain the embeddings $g_\phi(\mathcal{S}^j)$ of each instance $\mathcal{S}^j \in \mathcal{S}$ and the embedding $g_\phi(\mathcal{B}^j)$ of its circuit information $\mathcal{B}^j$. Then we utilize these embeddings to retrieve corresponding instances and circuits, respectively. The retriever is performed in two steps:

**Relevance:** We first identify the most relevant circuits based on their circuit information using $K$-Nearest Neighbors (KNN) search (Cover & Hart, 1967), a classical search technique. For a task involving a new circuit $\mathcal{B}^i$, we compute the cosine similarity between $g_\phi(\mathcal{B}^i)$ and all previously seen circuits $\mathcal{B}^j \in \mathcal{B}$, selecting the top-$k_1$ relevant circuits. To better align with real-world scenarios, note that typically $\mathcal{B}^i \notin \mathcal{B}$.

**Diversity:** Among the top-$k_1$ relevant circuits, diverse instances, such as excellent, poor, and failed designs, are all valuable for improving the new design. Excellent instances provide insights for optimal designs, while poor or failed instances help the model avoid ineffective model choices or hyperparameters. To capture this diversity, we apply spectral clustering (Ng et al., 2001), a technique

---

**Algorithm 1** LLM4Floorplan System

---

**Input:** Database $\mathcal{D} = (\mathcal{S}, \mathcal{B}, \mathcal{C})$, requirement document $\mathcal{R}^j$, current circuit $\mathcal{B}^j$, number of iterations $iters$, the number of relevant circuits $k_1$, the number of clusters $k_2$.
**Output:** Updated instance set $\mathcal{S}$ and circuit set $\mathcal{B}$.
 1: Initialize information state $\mathcal{A}_1 = \{\mathcal{R}^j, \mathcal{B}^j\}$;
 2: $\boldsymbol{r}_1 = \text{RetrieveInstances}(\mathcal{S}, \mathcal{B}, \mathcal{B}_j, k_1, k_2)$;      ▷ Retrieve instances based on Sec. 4.1.
 3: **for** $j = 1$ to $iters$ **do**
 4:     $\hat{\boldsymbol{t}}_1 = \mathcal{C}^j \sim f_{\boldsymbol{\theta}}(\cdot|\mathcal{P}_1, \boldsymbol{r}_1, \mathcal{A}_1)$;                     ▷ Generate model.
 5:     $\mathcal{A}_2 = \mathcal{A}_1 \cup \{\mathcal{C}^j\}$;                 ▷ Add the generated model to the state.
 6:     $\boldsymbol{r}_2 = \text{RetriveParameters}(\hat{\boldsymbol{t}}_1)$;           ▷ Retrieve parameter files based on the model.
 7:     $\hat{\boldsymbol{t}}_2 = \mathcal{H}^j \sim f_{\boldsymbol{\theta}}(\cdot|\mathcal{P}_{1:2}, \boldsymbol{r}_{1:2}, \hat{\boldsymbol{t}}_1, \mathcal{A}_2)$;                ▷ Generate hyperparameters.
 8:     $\mathcal{A}_3 = \mathcal{A}_2 \cup \{\mathcal{H}^j\}$;                     ▷ Add hyperparameters to the state.
 9:     $\boldsymbol{r}_3 = \text{RetrieveCode}(\hat{\boldsymbol{t}}_{1:2})$;        ▷ Retrieve code based on model and hyperparameters.
10:     $\hat{\boldsymbol{t}}_3 \sim f_{\boldsymbol{\theta}}(\cdot|\mathcal{P}_{1:3}, \boldsymbol{r}_{1:3}, \hat{\boldsymbol{t}}_{1:2}, \mathcal{A}_3)$;                     ▷ Generate new code.
11:     $\boldsymbol{r}_4 = (\mathcal{M}^j, \mathcal{I}^j) = \text{ExecuteModel}(\hat{\boldsymbol{t}}_{1:3})$;        ▷ Execute the model and obtain results.
12:     $\mathcal{A}_4 = \mathcal{A}_3 \cup \{\mathcal{M}^j\}, \mathcal{A}^j \triangleq \mathcal{A}_4$;                     ▷ Add evaluation metrics.
13:     $\mathcal{N}^j \sim h_{\boldsymbol{\psi}}(\cdot|\mathcal{P}_{\mathcal{N}}, \mathcal{A}^j, \mathcal{I}^j)$;                 ▷ Generate comprehensive comment.
14:     $\mathcal{S}^j \triangleq (\mathcal{A}^j, \mathcal{I}^j, \mathcal{N}^j)$;                     ▷ Construct new instance.
15:     $\mathcal{B} = \mathcal{B} \cup \{\mathcal{B}^j\}, \mathcal{S} = \mathcal{S} \cup \{\mathcal{S}^j\}$;             ▷ Add new circuit and instance to the sets.
16:     Add $\mathcal{S}^j$ to $\boldsymbol{r}_1$;
17: **end for**

---

well-suited for high-dimensional data, to the embeddings $g_{\boldsymbol{\phi}}(\mathcal{S}^j)$ of all instances $\mathcal{S}^j \in \mathcal{S}$ within the top-$k_1$ relevant circuits, selecting $k_2$ instances that are respectively closest to the $k_2$ clusters.

These two classical techniques are simple, effective, and efficient to retrieve relevant and diverse instances to enhance the agent decisions.

### 4.2 DRAT Prompting

To address the second challenge, we introduce a Dynamic Retrieval-Augmented Thought (DRAT) prompting technique, which aims to extract case-specific design objectives. Unlike Eq. 2 in RAT (Wang et al., 2024d), where the general LLMs cannot directly interact with specific physical design models and therefore cannot generate the corresponding model outputs, it becomes intractable to generate valid *zero-shot* thought steps $\mathcal{T}$. Thus, we maintain a dynamic state process $\mathcal{A}_{1:T}$ that is updated synchronously with the thought steps and split the task prompt $\mathcal{P}$ into separate ones $\mathcal{P} = \{\mathcal{P}_i\}_{i=1}^{T}$ to correspond each thought $\hat{\boldsymbol{t}}_i$ to a prompt $\mathcal{P}_i$. Consequently, we revise Eq. 2 from thought revision to a dynamic thought reasoning process:

$$\hat{\boldsymbol{t}}_1 \sim f_{\boldsymbol{\theta}}(\cdot|\mathcal{P}_1, \boldsymbol{r}_1, \mathcal{A}_1), \quad \hat{\boldsymbol{t}}_i \sim f_{\boldsymbol{\theta}}(\cdot|\mathcal{P}_{1:i}, \hat{\boldsymbol{t}}_{1:(i-1)}, \boldsymbol{r}_{1:i}, \mathcal{A}_i)\ (2 \le i \le T), \quad (3)$$

where $\boldsymbol{r}_i$ is relevant documents retrieved by the query $\boldsymbol{q}_i = g_{\boldsymbol{\phi}}(\hat{\boldsymbol{t}}_{1:(i-1)})$. The final step $\mathcal{A}_T$ is equal to the information set $\mathcal{A}^j$ of the $j$-th instance, which we will detail in Sec. 4.3. This dynamic state process is mandatory in our scheme as the subsequent decision is dependent of the previous ones.

### 4.3 LLM4Floorplan System

In this section, we present the LLM4Floorplan system, utilizing the search-cluster-based retriever introduced in Sec. 4.1 and the DRAT prompting described in Sec. 4.2. The corresponding algorithm is provided in Alg. 1. While this reasoning process is intricate, it fully adheres to Eq. 3 and serves as a general application that simulates the design workflow of human engineers.

To begin with, we define each instance as $\mathcal{S}^j = (\mathcal{A}^j, \mathcal{I}^j, \mathcal{N}^j) \in \mathcal{S}$ in the database, where $\mathcal{A}^j = \{\mathcal{R}^j, \mathcal{B}^j, \mathcal{C}^j, \mathcal{H}^j, \mathcal{M}^j\}$ is an information set includes $\mathcal{S}^j$'s requirement document $\mathcal{R}^j$, basic circuit information $\mathcal{B}^j$, model choice $\mathcal{C}^j$, hyperparameters $\mathcal{H}^j$, and metrics $\mathcal{M}^j$, while $\mathcal{I}^j$ is the result layout image and $\mathcal{N}^j \sim h_{\boldsymbol{\psi}}(\cdot|\mathcal{P}_{\mathcal{N}}, \mathcal{A}^j, \mathcal{I}^j)$ with parameters $\psi$ represents the comment that is generated by multi-modal LLM given $\mathcal{A}^j$ and $\mathcal{I}^j$. Then, LLM4Floorplan makes a chain of decisions following Eq. 3, containing the following components:

**Retrival System $r_{1:4}$:** The retrieval output $r_1$ is a set of instances that are retrieved based on the search-cluster-based retriever in Sec. 4.1. $r_2$ is the parameter files retrieved based on the model while $r_3$ is the code files retrieved based on the model and parameters. $r_4$ is a tuple of output including metrics and output image retrieved by executing the selected model with corresponding parameters and revised code.

**Agent Decisions $\hat{t}_{1:3}$ and $\mathcal{N}^j$:** Four decisions are generated by LLM agents, including model, hyperparameter, code, and comment generation, which encompass most design steps in real-world scenarios. These decisions are highlighted in darkblue in Alg. 1. Note that the first three decisions leverage the same LLM backbone $f_\theta$, as they involve purely textual data (including code), while the final decision utilizes a multi-modal LLM $h_\psi$, as it requires image input.

**Dynamic State Process $\mathcal{A}_{1:4}$:** The process from $\mathcal{A}_1$ to $\mathcal{A}_4$ represents the filling of information. Specifically, at first, $\mathcal{A}_1$ only has the requirement document $\mathcal{R}^j$ and the circuit information $\mathcal{B}^j$ for the $j$-th instance. With LLMs generating the model $\mathcal{C}^j$, hyperparameters $\mathcal{H}^j$, and the output metrics $\mathcal{M}^j$ step-by-step, $\mathcal{A}$ is finally obtained and is incorporated into the new instance $\mathcal{S}^j$.

Additionally, LLM4Floorplan incorporates $iters$, the number of iterations that simulates human engineers to fine-tune the model, hyperparameters, and codes based on the same circuit and requirement to obtain a better result. We show the whole decision prompting in Appendix C.1 and the comment prompting in Appendix C.2.

## 5 EXPERIMENT

This section outlines the experimental protocols in Sec. 5.1. We compare LLM4Floorplan's performance on public datasets (Sec. 5.2) and introduce a new benchmark with six distinct design requirements (Sec. 5.3). Further analysis in terms of the effect of code revision, search-cluster-based retriever, and ablation studies are shown in Sec. 5.4. Experiments are run on a machine with an AMD EPYC 7402 24-Core Processor, an NVIDIA GeForce RTX 4090, and 512GB RAM, repeated three times with different seeds, reporting the best result.

### 5.1 EXPERIMENTAL PROTOCOLS

**Datasets.** We incorporate two public datasets, **GSRC**[1] and **MCNC**[2], which are widely-used in floorplan. GSRC contains six circuits with number of blocks ranging from 10 to 300 while MCNC contains two circuits named *ami33* and *ami49*. A brief summary of these circuits is shown in Appendix D.1. Note that the scale of the largest circuit *n300* in MCNC is significantly larger than the ones of most industrial circuits, as stated in (Mallappa et al., 2024).

**Metrics.** Following previous floorplan methods (Li et al., 2022), we utilize the total HPWL, as defined in Eq. 1, calculated across all nets. To evaluate the agents' comprehension of design requirements, we introduce the Success Rate (SR), which quantifies the proportion of cases that meet the Overlapping Ratio (OR) criteria. For a comprehensive performance assessment, we propose the Rank metric, primarily based on SR. A higher SR guarantees a better Rank, and when SR values are identical, a lower HPWL results in an improved Rank. Further details on the total HPWL and OR calculations are provided in Appendix D.2.

**Floorplan Backbones.** We leverage two typical types of floorplan approaches: 1) PeF (Li et al., 2022), which is a representative analytical approach and is also the current *state-of-the-art*; 2) ECS (Chiou et al., 2016), which is a simulated annealing (SA)-based approach using the corner sequence representation. Compared to analytical approaches, ECS is capable of eliminating the overlap area without a second stage named legalization (Moffitt et al., 2006; Lin et al., 2016; Kai et al., 2023). Legalization permits the floorplan not necessarily non-overlapping but the overlap ratio cannot be too large. In our experiments, we regard the non-overlap as an advantage of ECS and retain the disadvantage of overlap area for PeF to evaluate the capability of LLMs to choose models under different requirements.

**LLM Backbones.** We employ multiple LLMs as agent backbones to implement $f_\theta$ introduced in Sec. 4.2, including GPT-3.5-turbo (Brown, 2020), GPT-4-turbo (Achiam et al., 2023), GPT-4o-

---

[1] http://vlsicad.eecs.umich.edu/BK/GSRCbench/
[2] http://vlsicad.eecs.umich.edu/BK/MCNCbench/

Table 1: Main floorplan results on eight circuits with various LLM backbones using DRAT prompting and the full LLM4Floorplan. Results that do not meet the OR ≤ 5% criterion are in gray, and failed results are marked as 'N/A'. The best results for each approach are highlighted in cyan.

| Method | LLM Backbones | Total HPWL of Circuits | | | | | | | | Overall Metrics | | |
|---|---|---|---|---|---|---|---|---|---|---|---|---|
| | | n10 | n30 | n50 | n100 | n200 | n300 | ami33 | ami49 | SR*↑ | WLR*↓ | Rank*↓ |
| | | | | | | **Analytical Approach** | | | | | | |
| PeF (Li et al., 2022) | | 37,097 | 104,488 | 130,589 | 198,685 | 361,313 | 480,571 | 59,061 | 725,235 | 1.000 | 1.000 | 6 |
| DRAT | DeepSeek-Chat | 35,797 | 102,152 | 126,336 | 195,268 | 354,605 | 473,402 | 61,981 | 745,263 | 1.000 | 0.992 | 5 |
| | DeepSeek-Coder | 39,201 | 102,650 | 133,187 | 195,713 | 379,148 | 481,584 | 61,120 | 737,008 | 1.000 | 1.018 | 8 |
| | GPT-3.5 | 38,464 | 106,658 | 126,670 | 194,066 | 369,972 | 494,165 | 58,192 | 733,926 | 1.000 | 1.007 | 7 |
| | GPT-4 | 34,967 | 99,652 | 124,546 | 189,354 | 346,186 | 456,867 | 60,194 | 708,398 | 0.625 | 0.956 | 10 |
| | GPT-4o-mini | 35,896 | 99,730 | 126,379 | 194,375 | 367,964 | 474,257 | 57,673 | 710,099 | 1.000 | 0.979 | 3 |
| | GPT-4o | 34,937 | 99,620 | 123,792 | 193,461 | 340,055 | 458,696 | 56,521 | 674,498 | 0.500 | 0.949 | 11 |
| | Claude-3.5 | 35,479 | 99,777 | 125,941 | 190,482 | 348,836 | 464,595 | 56,330 | 669,894 | 0.750 | 0.956 | 9 |
| LLM4Floorplan | GPT-4o-mini | 35,589 | 102,444 | 127,492 | 193,986 | 353,332 | 478,474 | 63,378 | 700,512 | 1.000 | 0.988 | 4 |
| | GPT-4o | 34,907 | 99,704 | 124,116 | 196,169 | 353,526 | 469,933 | 58,663 | 700,512 | 1.000 | 0.969 | 2 |
| | Claude-3.5 | 34,966 | 100,071 | 124,701 | 190,605 | 357,754 | 469,875 | 55,795 | 675,684 | 1.000 | 0.957 | 1 |
| | | | | | | **Simulated Annealing (SA)-based Approach** | | | | | | |
| ECS (Chiou et al., 2016) | | 40,082 | 123,022 | 168,848 | 295,387 | 561,956 | 848,366 | 82,454 | 1,445,688 | 1.000 | 1.000 | 4 |
| DRAT | DeepSeek-Chat | N/A | N/A | N/A | N/A | N/A | N/A | N/A | N/A | 0.000 | N/A | N/A |
| | DeepSeek-Coder | N/A | N/A | N/A | N/A | N/A | N/A | N/A | N/A | 0.000 | N/A | N/A |
| | GPT-3.5 | N/A | 123,102 | 172,105 | N/A | N/A | N/A | N/A | N/A | 0.250 | 1.010 | 5 |
| | GPT-4 | N/A | N/A | N/A | N/A | N/A | N/A | N/A | N/A | 0.000 | N/A | N/A |
| | GPT-4o-mini | N/A | N/A | N/A | N/A | N/A | N/A | 76,104 | N/A | 0.125 | 0.982 | 6 |
| | GPT-4o | N/A | N/A | N/A | N/A | N/A | 839,226 | N/A | N/A | 0.125 | 0.989 | 7 |
| | Claude-3.5 | N/A | N/A | N/A | N/A | N/A | N/A | N/A | N/A | 0.000 | N/A | N/A |
| LLM4Floorplan | GPT-4o-mini | 39,130 | 126,168 | 168,198 | 283,538 | 568,540 | 863,628 | 70,769 | 1,442,816 | 1.000 | 0.981 | 3 |
| | GPT-4o | 35,660 | 120,142 | 164,987 | 280,063 | 553,580 | 829,065 | 79,417 | 1,462,553 | 1.000 | 0.966 | 1 |
| | Claude-3.5 | 37,086 | 118,800 | 166,750 | 284,426 | 551,714 | 832,472 | 83,730 | 1,400,434 | 1.000 | 0.974 | 2 |

\* **SR**: Success rate; **WLR**: Average wirelength ratio compared to baselines (PeF Li et al. (2022) & ECS (Chiou et al., 2016)); **Rank**: Rank is determined primarily by SR, with a larger SR ensuring a better Rank. Within the same SR, a smaller WLR leads to a better Rank.

\# Rand(n) represents selecting $n$ random instances in Sec. 4.1;

mini (OpenAI, 2024b), GPT-4o (OpenAI, 2024a), Claude-3.5-Sonnet (Anthropic, 2024), DeepSeek-Chat & DeepSeek-Coder (Liu et al., 2024). We also exploit the same backbone for comment agent $h_\psi$; however, when it is not multi-modal LLM, e.g., GPT-3.5-turbo, we use the GPT-4o-mini instead regarding its high cost-effectiveness.

**Other settings.** For the text encoder $g_\phi$ introduced in Sec. 4.2, we simply use bge-small-en-v1.5 (Xiao et al., 2023), which is proved to be effective and efficient in text encoding with more reasonable similarity distribution. Additionally, we set $k_1 = 3$, $k_2 = 10$, and $iters = 3$, which are introduced in Sec. 4.1 and Sec. 4.3. Corresponding ablation studies can be referred to in Sec. 5.4. The construction of the database introduced in Sec. 4.1 is shown in Appendix D.4. The default parameters of baselines PeF (Li et al., 2022) and ECS (Chiou et al., 2016) are shown in Appendix D.5.

## 5.2 MAIN RESULTS

We present the comparisons of total HPWL for each circuit, as well as the success rate (SR), average wirelength ratio (WLR) relative to the baselines, and Rank in Table 1, using two floorplan backbones, PeF (Li et al., 2022) and ECS (Chiou et al., 2016), along with several LLM backbones. Results with an overlapping ratio exceeding 5%, a simple criterion, are considered unsatisfactory, which are marked in gray in Table 1. Failed results are indicated as 'N/A' [3]. Note that we were unable to obtain reasonable results using merely CoT (Wei et al., 2022) or RAT (Wang et al., 2024d). Furthermore, applying other agent-based baselines, such as flow-level agents (Liu et al., 2023a; Wu et al., 2024), is infeasible, as we are the first to introduce a task-level agent in physical design. However, even without the retriever introduced in Sec. 4.1, our method, LLM4Floorplan (w/o retriever), i.e., pure DRAT prompting, can still produce workable results, making it a strong baseline. Additionally, we use GPT-4o-mini, GPT-4o, and Claude-3.5, which are empirically more aggressive in achieving better HPWL, as backbones for the full version of LLM4Floorplan. Note that we do not employ GPT-4, as it is very similar to GPT-4o in practice but significantly more resource-consuming.

From an overall perspective in Table 1, for the analytical approach, DRAT with DeepSeek-Chat, DeepSeek-Coder, and GPT-3.5 backbones achieve all successful results but the WLR is almost the same as that of PeF (Li et al., 2022), which potentially indicates that they highly follow the origi-

---

[3]This usually occurs when the floorplan region is too small, preventing the simulated annealing (SA)-based approach from finding a solution that places all modules without overlap.

Table 2: Illustration of six distinct requirements with different optimization objectives, model pools for selection, and the need for code revisions.

| Index | Type | Objective | Model Selection | Code Revision |
|-------|------|-----------|-----------------|---------------|
| Req. 1 | Simple | $\min_\Theta \sum_{e_i \in \mathcal{E}} \text{HPWL}(e_i)$, s.t. OR $\leq 0.05$ | {Analytical} | ✗ |
| Req. 2 | Simple | $\min_\Theta \sum_{e_i \in \mathcal{E}} \text{HPWL}(e_i)$, s.t. OR $\leq 0.05$ | {SA} | ✗ |
| Req. 3 | Moderate | $\min_\Theta \sum_{e_i \in \mathcal{E}} \text{HPWL}(e_i)$, s.t. OR $\leq 0.025$ | {Analytical, SA} | ✗ |
| Req. 4 | Strict | $\min_\Theta \sum_{e_i \in \mathcal{E}} \text{HPWL}(e_i)$, s.t. OR $\leq 0.01$ | {Analytical, SA} | ✗ |
| Req. 5 | Specialized | $\min_\Theta \sum_{e_i \in \mathcal{E}} \text{HPWL}(e_i) \times (1 + 10 \times \text{OR})$ | {Analytical, SA} | ✗ |
| Req. 6 | Specialized | $\min_\Theta \sum_{e_i \in \mathcal{E}} \text{HPWL}(e_i)$, s.t. 1) OR $\leq 0.075$ 2) blocks $b_1$ and $b_2$ are adjacent. | {Analytical, SA} | ✓ |

nal PeF. On the contrary, DRAT with GPT-4, GPT-4o, and Claude-3.5 backbones are aggressive to achieve better WLR but bring about the sacrifice of SR. An exception is DRAT with GPT-4o-mini, which effectively balance the SR and WLR. This phenomenon implies that DRAT with powerful LLMs, e.g., GPT-4, GPT-4o, and Claude-3.5, might be even worse than other moderate LLMs as these powerful LLMs integrate too much case-specific guidance but ignore the domain-specific expertise. However, equipped with DRAT and retriever, LLM4Floorplan (full version) with GPT-4o-mini, GPT-4o, and Claude-3.5 backbones maintain the 100% SR while keeping low WLR.

The advantage of LLM4Floorplan becomes even more pronounced compared to the SA-based approach, where DRAT with all backbones fails in almost all circuits. This is mainly due to the fact that the floorplan region is too small, preventing the approach from finding a suitable solution that accommodates all modules without overlap. However, with the retriever, LLM4Floorplan using the GPT-4o-mini, GPT-4o, and Claude-3.5-Sonnet backbones achieves a 100% SR while maintaining a lower WLR than ECS (Chiou et al., 2016).

An interesting question arises: *Can LLM4Floorplan maintain its superiority when faced with stricter OR criteria or even other types of requirements?* To address this, we propose a benchmark in Sec. 5.3 with six distinct floorplan requirements and evaluate the performance of LLM4Floorplan.

## 5.3 Benchmark and Meeting Diverse Requirements

We evaluate LLM4Floorplan against diverse requirements by introducing a novel benchmark comprising six distinct criteria. These criteria span empirically simple, moderate, strict, and specialized requirements, as summarized in Table 2, with varying objectives, model pools for selection, and the need for code revisions. Further details of the requirements are provided in Appendix D.3. Note that the capabilities of hyperparameter tuning and performance evaluation are incorporated for all requirements. The first two requirements correspond to the results in the main results in Table 1.

As shown in Table 3, LLM4Floorplan achieves the highest rank in the first five requirements. Notably, for the strict OR criteria (Req. 4), LLM4Floorplan with GPT-4o-mini, GPT-4o, and Claude-3.5 backbones significantly improves the SR from an average of 0.250 to 0.875, compared to DRAT. An interesting observation is that DRAT with the GPT-3.5 backbone, which is generally considered less powerful, achieves an SR of 1.000 in Req. 4. This is primarily because it tends to choose SA approach to avoid overlapping areas under strict OR criteria; however, this comes at a substantial cost to WLR and remains inferior to LLM4Floorplan with the Claude-3.5 backbone. The only failure of LLM4Floorplan occurs in Req. 6, where there is no notable improvement in SR. This is latently due to the absence of relevant instances in the database for the retriever to locate, preventing LLM4Floorplan from learning any useful guidance. Nonetheless, with a powerful LLM backbone (e.g., Claude-3.5), our pure DRAT is still capable of handling difficult and unforeseen design requirements. Further details of Table 3 are provided in Appendix E.3.

## 5.4 Futher Analysis

**Code Revision.** One promising finding is that LLM4Floorplan can comprehend unseen requirements (e.g., Req. 6 in Table 2) and generate revised code to achieve the corresponding objectives. This capability is not only attributed to its understanding of requirement documents but also to

```
Code Revision

# Original Code                              # Revised Code
def forward(self, density_weight):          def forward(self, density_weight):
    ...                                          ...
    loss = self.ana_loss(density_coef)          loss = self.ana_loss(density_coef)
    return loss                                 return loss

def ana_loss(self, density_coef):           def ana_loss(self, density_coef):
    loss = self.hpwl + self.energy * density_coef    adjacency_penalty = self.calc_adjacency_penalty()
    return loss                                 overlap_penalty = self.calc_overlap_penalty()
                                                loss = self.hpwl + self.energy * density_coef +
┌──────────────────────────────────────┐            adjacency_penalty + overlap_penalty
│ Revisions Made by the LLM:            │       return loss
│ 1) LLM adds calc_overlap_penalty      │
│ function to address the first constraint, ensur- │ def calc_adjacency_penalty(self):
│ ing that overlap ratio (OR) remains below 5%. │     first_block = self.block_positions[0]
│ 2) LLM adds calc_adjacency_penalty    │       second_block = self.block_positions[1]
│ function satisfy the second constraint, which │   distance = torch.norm(first_block - second_block)
│ requires that the first two blocks be adjacent. │ return 1000 * distance  # Adjust the weight as needed
└──────────────────────────────────────┘
                                            def calc_overlap_penalty(self):
                                                overlap_threshold = 0.075
                                                penalty = torch.relu(self.overlap - overlap_threshold)
                                                return 10000 * penalty  # Adjust the weight as needed
```

Figure 4: Example of code revision by LLMs. The original code is shown on the left, while the revised version, highlighted in teal, is on the right. Incorporated with DRAT, LLM4Floorplan is able to understand the requirements and automatically add two corresponding loss functions.

Table 3: Floorplanning results for six distinct requirements. The backbones and circuits are the same as those in Table 1. Failed results are marked as 'N/A'. The top-ranked result for each requirement is highlighted in cyan.

| Method | LLM Backbones | Req. 1 | | Req. 2 | | Req. 3 | | Req. 4 | | Req. 5 | Req. 6 | |
|---|---|---|---|---|---|---|---|---|---|---|---|---|
| | | SR*↑ | WLR*↓ | SR*↑ | WLR*↓ | SR*↑ | WLR*↓ | SR*↑ | WLR*↓ | Obj. VR#↓ | SR*↑ | WLR*↓ |
| PeF (Li et al., 2022) | | 1.000 | 1.000 | / | / | 0.750 | 1.000 | 0.000 | 1.000 | 1.000 | 0.000 | 1.000 |
| ECS (Chiou et al., 2016) | | / | / | 1.000 | 1.000 | 1.000 | 1.463 | 1.000 | 1.462 | 1.190 | 0.000 | 1.469 |
| DRAT | DeepSeek-Chat | 1.000 | 0.992 | 0.000 | N/A | 0.625 | 0.996 | 0.625 | 1.044 | 0.993 | 0.000 | 0.999 |
| | DeepSeek-Coder | 1.000 | 1.018 | 0.000 | N/A | 0.875 | 0.988 | 0.625 | 1.044 | 0.963 | 0.000 | 0.973 |
| | GPT-3.5 | 1.000 | 1.007 | 0.250 | 1.010 | 0.750 | 1.006 | 1.000 | 1.281 | 1.130 | 0.000 | 1.014 |
| | GPT-4 | 0.625 | 0.956 | 0.000 | N/A | 0.125 | 0.985 | 0.250 | 1.025 | 0.923 | 0.125 | 0.952 |
| | GPT-4o-mini | 1.000 | 0.979 | 0.125 | 0.982 | 0.875 | 0.991 | 0.250 | 1.020 | 0.933 | 0.250 | 0.989 |
| | GPT-4o | 0.500 | 0.949 | 0.125 | 0.989 | 0.375 | 0.970 | 0.000 | 1.031 | 0.984 | 0.125 | 0.969 |
| | Claude-3.5 | 0.750 | 0.956 | 0.000 | N/A | 1.000 | 1.023 | 0.500 | 1.039 | 0.942 | 1.000 | 1.053 |
| LLM4Floorplan | GPT-4o-mini | 1.000 | 0.988 | 1.000 | 0.981 | 0.750 | 1.016 | 1.000 | 1.058 | 0.926 | 0.125 | 1.161 |
| | GPT-4o | 1.000 | 0.969 | 1.000 | 0.966 | 0.875 | 1.009 | 0.750 | 1.098 | 0.952 | 0.375 | 0.976 |
| | Claude-3.5 | 1.000 | 0.957 | 1.000 | 0.974 | 1.000 | 0.999 | 0.875 | 1.055 | 0.905 | 0.875 | 0.984 |

\* **SR**: Success rate; **WLR**: Average wirelength ratio compared to PeF Li et al. (2022) (Req. 1, Req. 3-6) and ECS (Chiou et al., 2016) (Req. 2); **Rank**: Rank is determined primarily by SR, with a larger SR ensuring a better Rank. Within the same SR, a smaller WLR leads to a better Rank.
\# Objective value ratio is compared with PeF Li et al. (2022). The objective value is defined as $\sum_{e_i \in \mathcal{E}} \text{HPWL}(e_i) \times (1 + 10 \times \text{OR})$, as described in Table 2.

our abstraction of the model's code. Specifically, as LLMs can experience reduced effectiveness when processing long contexts (Zhang et al., 2024), we isolate the core components of the floorplanning model, focusing on the main model function and loss functions. This design empirically reduces the complexity of the code, making it easier to understand. During interactions with these simplified code structures, LLMs are able to devise novel loss functions tailored to the given requirements. A successful example of code revision is shown in Fig. 4, where two functions, calc_overlap_penalty and calc_adjacency_penalty, are added to the original code to meet the specialized requirement. We present four cases in Fig. 5. The left two images correspond to the results of circuit n10, while the right two images show the results of circuit n100. In each pair, the left image illustrates the desirable outcome where the first two blocks in the block list ($b_0$ and $b_1$) are adjacent, whereas the right image displays the blocks as separate.

**Selected instances in Retriever.** We explore the effectiveness of the retriever introduced in Sec. 4.1. As illustrated in Fig. 6, Principal Component Analysis (PCA) (Abdi & Williams, 2010) is applied to reduce the dimensionality of the text file embeddings for all instances to 2-D, allowing us to visualize them in the reduced space. From a broader perspective, the instances are divided into three main types: those obtained by the analytical approach and the successful/failed cases from the simulated annealing (SA) approach. In Fig. 6(a), the 10 randomly selected instances fail to cover the type associated with the SA approach, providing no guidance for LLMs regarding SA. In contrast, Fig. 6(b) shows that, even with only 5 instances, all three types are represented. Furthermore, Fig. 6(c) shows that, as the number of selected instances increases according to the clusters, more

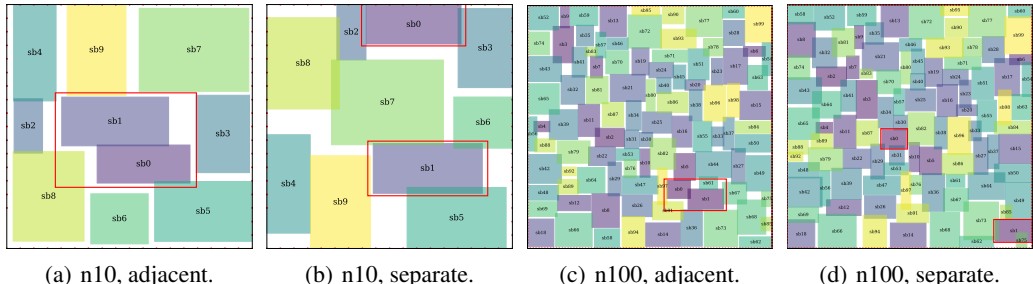

|       |       |       |       |
|-------|-------|-------|-------|
| (a) n10, adjacent. | (b) n10, separate. | (c) n100, adjacent. | (d) n100, separate. |

Figure 5: Demonstration of floorplanning for Req. 6 in Table 2. The left pair shows results for the n10 circuit, and the right pair for the n100 circuit. In each pair, the left image meets the requirement that the first two blocks (outlined in red) are adjacent, while the right does not.

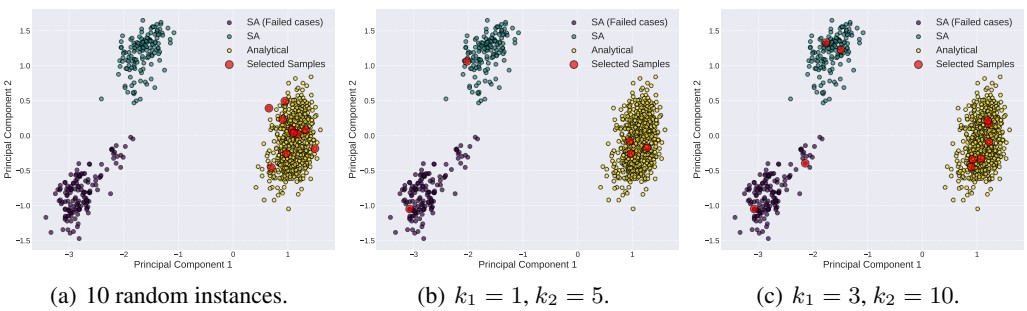

|       |       |       |
|-------|-------|-------|
| (a) 10 random instances. | (b) $k_1 = 1$, $k_2 = 5$. | (c) $k_1 = 3$, $k_2 = 10$. |

Figure 6: Scatters of the embeddings of the instances. a) Retrieve 10 instances randomly; b) Retrieve 5 instances corresponding to $k_1 = 1$, $k_2 = 5$ in the search-cluster-based retriever; c) Retrieve 10 instances according to $k_1 = 3$, $k_2 = 10$ in the search-cluster-based retriever.

representative instances are retrieved, enhancing the agent's comprehension. To illustrate in detail, we conduct the following ablation studies.

**Ablation Studies.** We evaluate the hyperparameters $k_1 \in \{1, 3\}$, $k_2 \in \{5, 10\}$ in Sec. 4.1, and $iters \in \{0, 3\}$ in Sec. 4.3. It is also crucial to evaluate the impact of integrating the model evaluation agent $h_\phi$. The best configuration is achieved with hyperparameters $k_1 = 3$, $k_2 = 10$, $iters = 3$, and with the inclusion of $h_\phi$. Detailed empirical results are presented in Appendix E.1.

## 6 CONCLUSION AND OUTLOOK

In this paper, we presented LLM4Floorplan, the first task-level multi-agent system for physical design, particularly targeting the floorplanning stage in EDA. By introducing a search-cluster-based retriever and the Dynamic Retrieval-Augmented Thought (DRAT) prompting technique, we addressed the challenges of domain-specific expertise integration and case-specific design guidance. Through extensive experiments on public datasets, we demonstrated the efficacy of LLM4Floorplan across diverse and complex requirements. The system achieved significant improvements in task comprehension, model selection, hyperparameter tuning, and performance evaluation. Our findings indicate that LLM4Floorplan successfully simulates the workflow of human engineers and provides strong generalization capabilities, paving the way for further research in applying LLM-powered agents to other stages of physical design.

This paper also has some *limitations* that suggest avenues for future work: 1) LLM4Floorplan faces challenges in handling highly novel designs due to its reliance on a predefined design database. 2) The system has yet to be evaluated on expanded benchmarks or integrated into other physical design stages, such as placement and routing.

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

# A   NOTATION

**Floorplan**

| | |
|---|---|
| $W$ | Width of the chip layout region. |
| $H$ | Height of the chip layout region. |
| $\mathcal{V}_b$ | Set of blocks. |
| $\mathcal{V}_p$ | Set of pins. |
| $\mathcal{E}$ | Set of nets. |
| $b_i$ | The $i$-th block in $\mathcal{V}_b$. |
| $p_i$ | The $i$-th pin in $\mathcal{V}_p$. |
| $e_i$ | The $i$-th net in $\mathcal{E}$. |
| $n_b$ | Number of blocks. |
| $n_p$ | Number of pins. |
| $n_e$ | Number of nets. |
| $x_{b_i}, y_{b_i}$ | Center coordinates of block $b_i$ along the x- and y-axes. |
| $x_{p_i}, y_{p_i}$ | Coordinates of pin $p_i$ along the x- and y-axes. |

**RAT**

| | |
|---|---|
| $\mathcal{D}$ | Standard database. |
| $\mathcal{S}$ | Set of running instances. |
| $\mathcal{B}$ | Set of basic circuit information. |
| $\mathcal{C}$ | Set of models. |
| $\mathcal{S}^j$ | The $j$-th instance in the set $\mathcal{S}$. |
| $\mathcal{B}^j$ | Basic circuit information corresponding to the $j$-th instance. |
| $\mathcal{C}^j$ | Model corresponding to the $j$-th instance. |
| $\mathcal{P}$ | Set of prompt steps. |
| $\mathcal{P}_i$ | The $i$-th prompt step. |
| $\mathcal{A}_i$ | The $i$-th state step. |
| $\hat{t}i$ | The $i$-th generated thought step. |
| $T$ | Number of thought steps. |
| $f_{\boldsymbol{\theta}}$ | Pretrained LLM with parameters $\boldsymbol{\theta}$. |
| $g_{\boldsymbol{\phi}}$ | Text encoder or LLM with pretrained parameters $\phi$. |
| $q_i$ | Query generated by $\hat{\boldsymbol{t}}_{1:(i-1)}$. |
| $\boldsymbol{r}_i$ | Relevant documents retrieved by query $q_i$. |
| $k_1$ | Number of relevant circuits in the retriever. |
| $k_2$ | Number of clusters in the retriever. |
| $\mathcal{A}^j$ | Information set corresponding to the $j$-th instance. |
| $\mathcal{I}^j$ | Output image corresponding to the $j$-th instance. |

| | |
|---|---|
| $\mathcal{N}^j$ | Comment corresponding to the $j$-th instance. |
| $\mathcal{R}^j$ | Requirement document corresponding to the $j$-th instance. |
| $\mathcal{H}^j$ | Hyperparameters corresponding to the $j$-th instance. |
| $\mathcal{M}^j$ | Metrics corresponding to the $j$-th instance. |
| $h_{\boldsymbol{\psi}}$ | Multi-modal LLM with parameters $\boldsymbol{\psi}$, used to generate comments. |

## B  DETAILS OF FLOORPLANNING PROBLEM

The vanilla fixed-outline floorplanning problem, described in Sec. 3, can be formulated as

$$\min_{\Theta} \quad \sum_{e_i \in \mathcal{E}} \text{HPWL}(e_i), \tag{4a}$$

$$s.t. \quad \text{no overlapping area among blocks}, \tag{4b}$$

$$\text{suitable aspect ratio}, \tag{4c}$$

$$\text{blocks are within the fixed outline}. \tag{4d}$$

Although the optimization variables $\Theta$ include the width $w_i$, height $h_i$, and coordinates $(x_i, y_i)$ of each block $b_i \in \mathcal{V}_b$, it is important to note that some parameters are fixed depending on the type of block:

- For *soft* blocks, width $w_i$, height $h_i$, and coordinates $(x_i, y_i)$ are all learnable.

- For *hard* blocks, width $w_i$ and height $h_i$ are fixed while coordinates $(x_i, y_i)$ are learnable.

- For *pre-placed* blocks, width $w_i$, height $h_i$, and coordinates $(x_i, y_i)$ are all fixed.

The first constraint in Eq. 4b ensures that there is no overlapping area among the blocks. The second constraint in Eq. 4c ensures that each block maintains an appropriate aspect ratio that satisfies the given criterion. For example, it may require the aspect ratio to lie between 1/3 and 3, meaning the width-to-height ratio must be within the range [1/3, 3]. The third constraint in Eq. 4d ensures that all blocks remain within the fixed outline.

For simulated annealing approaches, these constraints are easily maintained, though finding the optimal total HPWL can be challenging. For analytical approaches, the second and third constraints can be met by clipping the learned parameters. However, optimizing the first constraint to eliminate overlapping areas is more difficult. Therefore, in this paper, we examine the model's performance under varying overlapping ratio criteria.

## C  DOMAIN-SPECIFIC PROMPTS

### C.1  DECISION PROMPT

> **SYSTEM**
>
> You are an AI designer integrated with Electronic Design Automation (EDA) floorplanning. Your task is to decide the best approach, determine its parameters, and revise the key codes based on the requirement document, circuit information, previous results, parameter descriptions, and key codes.

---

**USER**

**Requirement Document:**
I want to minimize the wirelength as much as possible. Overlap should be less than 0.05, but minimizing wirelength is the primary objective. Any solutions that achieve a lower wirelength with slightly higher overlap (up to 0.05) are preferable over solutions with very low overlap but higher wirelength.

**Brief Circuit Information:**
```
{
    'circuit':  'n10',
    'num_blocks':  10,
    'num_pins':  69,
    'num_nets':  118
}
```

**Previous Results:**
```
{
    'exp0':  {'model':  ANALYTICAL, 'parameters':  [parameter H^0],
        'metrics':  [metric M^0], 'comment':  [comment N^0]},
    'exp1':  {'model':  SA, 'parameters':  [parameter H^1],
        'metrics':  [metric M^1], 'comment':  [comment N^1]},
    ...
}
```

**Instructions:**
IMPORTANT:
The previous results for the algorithm are described using the following format:
```
{
    'exp0': {'model': name, 'parameters': {'parameter1': value, ...}, 'metrics': {'metric1': value,
        ...}, 'comment': content},
    'exp1': {'model': name, 'parameters': {'parameter1': value, ...}, 'metrics': {'metric1': value,
        ...}, 'comment': content},
    ...
}
```
Each entry represents an experiment with its model, parameters, corresponding metrics, and comments. Please choose the most suitable model for the given circuit and requirements.

Note: The chosen model must be either the analytical model or the simulated annealing model. You must reply with ANALYTICAL or SA.
Respond directly to the instruction. DO NOT include additional explanations or introductions unless specifically asked to do so.

---

**ASSISTANT**

ANALYTICAL

---

**USER**

**Parameter Descriptions:**

```python
import argparse

def get_analytical_args(parser):
    parser.add_argument('grid_size', type=int, default=128, help='
        Control the granularity of floorplan region segmentation. A
         larger grid_size tends to reduce overlap but will take
        more time.')
    parser.add_argument('amplify_ratio', type=float, default=1.15,
        help='amplify_ratio must not be less than 1. It controls
        the area of the floorplan region. A larger amplify_ratio
        tends to reduce overlap but will increase wirelength.')
    parser.add_argument('init_lr', type=float, default=0.1, help='
        The initial learning rate. A too small init_lr will lead to
         underfitting, but a too large init_lr will also lead to
        nonconvergence.')
```

---

```
        parser.add_argument('density_weight', type=float, default
            =100000.0, help='The coefficient of density. A larger
            density weight tends to reduce overlap but will increase
            wirelength.')
        return parser
```

**Instructions:**
IMPORTANT:
Based on your experience and the provided parameter descriptions, choose suitable values for each parameter.
The parameters and metrics for the algorithm are described in the following format:
{
    'parameter1': value1,
    'parameter2': value2,
    ...
}

Note: The refined parameters do not have to be within the previous results.
Respond directly to the instruction. DO NOT include additional explanations or introductions unless specifically asked to do so.

## ASSISTANT

```
{
    'grid_size':  128,
    'amplify_ratio':  1.15,
    'init_lr':  0.1,
    'density_weight':  100000.0
}
```

## USER

**Key Codes:**

```
import numpy as np
import torch
import torch.nn as nn
from torch.nn import Parameter

import hpwl, density, potential
import move
import scheduler

class ANA(nn.Module):
    def __init__(self, floorplan, grid_size, white_space, device=
        torch.device('cuda:0')):
        super().__init__()

        ...

    def forward(self, density_weight):

        ...

        loss = self.ana_loss(density_coef)
        return loss

    def ana_loss(self, density_coef):
        loss = self.hpwl + self.energy * density_coef
        return loss
```

**Instructions:**
IMPORTANT:
Based on your experience and the provided key codes, you can either keep the current code or revise it by adding new objective functions to meet the requirement. If you keep the current code, just output 'KEEP'. Otherwise, please output the whole revised code.
Note: The revised code should be runnable and must contain the original function names.

Respond directly to the instruction.
DO NOT include additional explanations or introductions unless specifically asked to do so.

**ASSISTANT**

KEEP

## C.2 COMMENT PROMPT

**SYSTEM**

You are an AI designer integrated with Electronic Design Automation (EDA) floorplanning.
Based on the requirement document, brief circuit information, parameters, results (including metrics and the floorplan image), your task is to provide a specialized comment on them, specifically emphasizing:

1. How the final result meets the requirements outlined in the requirement document.
2. The impact and suggestion of various parameters on the final result.
3. The description of the layout image (if given).
Note: The answer should be within 300 tokens.

**USER**

**Requirement Document:**
I want to minimize the wirelength as much as possible. Overlap should be less than 0.05, but minimizing wirelength is the primary objective. Any solutions that achieve a lower wirelength with slightly higher overlap (up to 0.05) are preferable over solutions with very low overlap but higher wirelength.
**Brief Circuit Information:**
```
{
    'circuit':  'n10',
    'num_blocks':  10,
    'num_pins':  69,
    'num_nets':  118
}
```
**Parameter:**
```
{
    'grid_size':  128,
    'amplify_ratio':  1.15,
    'init_lr':  0.1,
    'density_weight':  100000.0
}
```
**Metric:**
```
{
    'wirelength':  37127.79,
    'overlap':  0.0136,
    'time':  8.16
}
```
**Image:** [image $\mathcal{I}^j$]

> **ASSISTANT**
>
> The final floorplan successfully aligns with the requirements outlined in the document, emphasizing minimal wirelength while adhering to the overlap constraint. With a wirelength of **37127.79** and an overlap of **0.0136**, the layout optimally meets the requirement of keeping overlap below the 0.05 threshold. This indicates a strong prioritization of wirelength minimization, demonstrating that small overlaps can be tolerated if they lead to more compact wiring.
>
> The selected parameters played a critical role in achieving these results. The **initial learning rate (init_lr)** of **0.1** likely facilitated rapid convergence during the optimization process, while the **amplify ratio** of **1.15** helped balance expansion and contraction movements of blocks. The high **density weight** effectively guided the layout towards minimizing wirelength by reducing congestion, greatly impacting the overall efficiency.
>
> The layout image depicted shows the spatial arrangement of various blocks (labeled **sb0 to sb9**), indicating a thoughtful placement strategy that balances proximity to reduce wiring needs. The positioning reflects calculated overlaps, which are strategically limited to meet requirements without sacrificing the essential goal of minimizing wirelength.
>
> Overall, this floorplan not only meets but prioritizes the defined objectives, demonstrating effective EDA practices. Further optimization could explore slightly increasing overlap if it significantly decreases wirelength in future iterations.

# D  EXPERIMENTAL PROTOCOLS

## D.1  DATASET

We show the number of blocks, pins, nets of each circuit in Table 6.

Table 6: Number of blocks, pins, and nets in datasets GSRC and MCNC.

| CIRCUIT | GSRC | | | | | | MCNC | |
|---|---|---|---|---|---|---|---|---|
| | n10 | n30 | n50 | n100 | n200 | n300 | ami33 | ami49 |
| #Blocks | 10 | 30 | 50 | 100 | 200 | 300 | 33 | 49 |
| #Pins | 69 | 212 | 209 | 334 | 564 | 569 | 40 | 22 |
| #Nets | 118 | 349 | 485 | 885 | 1,585 | 1,893 | 123 | 408 |

## D.2  METRICS

**Total HPWL.** The total HPWL is the simple summation of the HPWL of each net $e_i \in \mathcal{E}$, which is calculated as follows

$$\sum_{e_i \in E} \text{HPWL}(e_i). \tag{5}$$

**Overlapping Ratio.** Overlapping ratio refers to the cumulative area where two or more blocks physically overlap within the defined floorplan region. To calculate the overlapping ratio, we first define the boundary box of each block $b_i \in \mathcal{V}_b$ determined by its left, right, bottom, and top boundaries:

$$x_{b_i}^{\text{left}} = x_{b_i} - w_i/2, \quad x_{b_i}^{\text{right}} = x_{b_i} + w_i/2, \quad y_{b_i}^{\text{bottom}} = y_{b_i} - h_i/2, \quad y_{b_i}^{\text{top}} = y_{b_i} + h_i/2. \tag{6}$$

Then, we check the pairwise intersection for each pair of blocks $(b_i, b_j)$. They overlap if

$$\max(x_{b_i}^{\text{left}}, x_{b_j}^{\text{left}}) < \min(x_{b_i}^{\text{right}}, x_{b_j}^{\text{right}}) \quad \text{and} \quad \max(x_{b_i}^{\text{bottom}}, x_{b_j}^{\text{bottom}}) < \min(x_{b_i}^{\text{top}}, x_{b_j}^{\text{top}}). \tag{7}$$

For each overlapping pair $(b_i, b_j)$, calculate the area of intersection $\text{OverlapArea}(b_i, b_j)$ as:

$$\left(\min(x_{b_i}^{\text{right}}, x_{b_j}^{\text{right}}) - \max(x_{b_i}^{\text{left}}, x_{b_j}^{\text{left}})\right) \times \left(\min(x_{b_i}^{\text{top}}, x_{b_j}^{\text{top}}) - \max(x_{b_i}^{\text{bottom}}, x_{b_j}^{\text{bottom}})\right). \tag{8}$$

Finally, the overlapping ratio is computed as

$$\text{OR} = \frac{\sum_{i<j} \text{OverlapArea}(b_i, b_j)}{W \cdot H}. \tag{9}$$

### D.3 INTRODUCTION OF THE BENCHMARK

We devise six distinct requirements on floorplan as follows, which corresponds to the illustration shown in Table 2 in Sec. 5.3.

- **Requirement 1/2.** I want to minimize the wirelength as much as possible. Overlap should be less than 0.05, but minimizing wirelength is the primary objective. Any solutions that achieve a lower wirelength with slightly higher overlap (up to 0.05) are preferable over solutions with very low overlap but higher wirelength.

- **Requirement 3.** I want to minimize the wirelength as much as possible, with the strict constraint that the overlap must always be less than 0.025. Within this constraint, minimizing wirelength is the primary objective. Solutions that achieve the lowest wirelength while maintaining overlap below 0.025 are preferred.

- **Requirement 4.** I want to minimize the wirelength as much as possible, with the strict constraint that the overlap must always be less than 0.01. Within this constraint, minimizing wirelength is the primary objective. Solutions that achieve the lowest wirelength while maintaining overlap below 0.01 are preferred.

- **Requirement 5.** I want to minimize the wirelength*(1+overlap*10).

- **Requirement 6.** Please revise the code to minimize the wirelength as much as possible while adhering to two strict constraints: 1) the first two blocks in the block list must be adjacent; 2) the overlap must always be less than 0.075.

In this benchmark, we evaluate the agent's capability of handling different types of design objectives, such as meeting different levels of OR requirements (Req. 1 to Req. 4), comprehensive objective (Req. 5), and personalized objective (Req. 6). Note that these requirements are designed to reflect diverse real-world scenarios. The OR constraint is introduced to facilitate easier legalization, while the adjacency constraint is aimed at optimizing feedthrough (Koide et al., 1996). Additionally, we also evaluate the agent's capability of model selection and code revision.

### D.4 CONSTRUCT THE DATABASE.

In Sec. 4.1, we have introduced a standard database to store instances for the retriever to extract relevant and diverse instances that are useful for the given new case. In this section, we provide details in terms of how to construct such database.

We employ two floorplanning baselines, PeF (Li et al., 2022) and ECS (Chiou et al., 2016), on eight circuits and run multiple cases with varying hyperparameters:

For PeF (Li et al., 2022), we choose the following hyperparameters:

- `grid_size` $\in \{64, 128, 256\}$, which control the granularity of floorplan region segmentation. A larger `grid_size` tends to reduce overlap but will take more time.

- `amplify_ratio` $\in \{1.05, 1.1, 1.15, 1.2\}$, which controls the area of the floorplan region. A larger `amplify_ratio` tends to reduce overlap but will increase wirelength.

- `init_lr` $\in \{0.01, 0.1, 1.0\}$, which means the initial learning rate. A too small `init_lr` will lead to underfitting, but a too large `init_lr` will also lead to non-convergence.

- `density_weight` $\in \{10000, 100000\}$, which means the coefficient of density. A larger density weight tends to reduce overlap but will increase wirelength.

For ECS (Chiou et al., 2016), we choose the following hyperparameters:

- `amplifyRatio` $\in \{1.05, 1.1, 1.15, 1.2\}$, which controls the area of the floorplan region. A smaller `amplifyRatio` tends to reduce wirelength but might fail to find a solution and output 'Fail'. `amplifyRatio` must be larger than 1.

Table 7: Ablation study under the 4th requirement in Sec. 5.3 on the GSRC and MCNC benchmarks, using the GPT-4o-mini and PeF (Li et al., 2022) backbones. Results that do not satisfy the overlapping constraint are highlighted in gray, while the best result is indicated in cyan.

| Hyperparameter | | | | Total HPWL of Circuits | | | | | | | | Metric | | |
|---|---|---|---|---|---|---|---|---|---|---|---|---|---|---|
| $h_\psi$ | $k_1$ | $k_2$ | iters | n10 | n30 | n50 | n100 | n200 | n300 | ami33 | ami49 | SR*↑ | WLR*↓ | Rank*↓ |
| PeF (Li et al., 2022) | | | | 37,128 | 104,564 | 130,944 | 198,685 | 361,313 | 481,350 | 59,061 | 740,577 | 0.000 | 1.000 | 15 |
| ✗ | ✗ | ✗ | 0 | 38,593 | 105,016 | 134,501 | 199,757 | 370,255 | 495,216 | 602,23 | 750,794 | 0.250 | 1.020 | 12 |
| ✗ | / | Rand(5)# | 0 | 35,915 | 102,152 | 128,529 | 199,757 | 353,213 | 476,975 | 60,702 | 742,671 | 0.000 | 0.991 | 14 |
| ✗ | / | Rand(10)# | 0 | 35,831 | 103,524 | 134,391 | 195,891 | 355,548 | 488,113 | 61,120 | 754,171 | 0.000 | 1.002 | 16 |
| ✓ | / | Rand(5)# | 0 | 37,768 | 111,845 | 134,501 | 217,176 | 371,465 | 492,884 | 63,852 | 774,645 | 0.500 | 1.048 | 9 |
| ✓ | / | Rand(10)# | 0 | 37,753 | 106,369 | 134,587 | 212,138 | 369,665 | 503,191 | 61,690 | 732,117 | 0.375 | 1.029 | 11 |
| ✗ | 3 | 5 | 0 | 38,079 | 107,714 | 133,397 | 208,439 | 370,154 | 479,339 | 58,972 | 729,055 | 0.500 | 1.016 | 8 |
| ✓ | 3 | 5 | 0 | 39,440 | 106,695 | 142,544 | 227,832 | 410,955 | 874,842 | 59,773 | 720,401 | 0.875 | 1.157 | 5 |
| ✓ | 3 | 5 | 3 | 38,908 | 106,695 | 138,237 | 227,832 | 410,955 | 528,701 | 59,773 | 720,401 | 0.875 | 1.061 | 4 |
| ✗ | 3 | 10 | 0 | 37,730 | 107,795 | 134,144 | 199,757 | 362,485 | 491,965 | 60,702 | 700,787 | 0.000 | 1.010 | 17 |
| ✓ | 3 | 10 | 0 | 37,915 | 119,500 | 133,293 | 204,252 | 409,757 | 520,078 | 60,223 | 792,246 | 0.625 | 1.064 | 6 |
| ✓ | 3 | 10 | 3 | 37,915 | 110,982 | 138,298 | 208,069 | 385,422 | 514,722 | 63,745 | 787,709 | 1.000 | 1.058 | 1 |
| ✗ | 1 | 5 | 0 | 41,466 | 106,369 | 133,019 | 207,522 | 356,838 | 502,148 | 60,702 | 700,787 | 0.000 | 1.025 | 18 |
| ✓ | 1 | 5 | 0 | 37,877 | 107,347 | 134,479 | 208,124 | 384,202 | 546,797 | 61,524 | 772,775 | 0.500 | 1.051 | 10 |
| ✓ | 1 | 5 | 3 | 37,877 | 112,955 | 134,479 | 203,389 | 369,671 | 543,799 | 59,704 | 772,775 | 0.875 | 1.045 | 2 |
| ✗ | 1 | 10 | 0 | 37,745 | 107,347 | 132,024 | 195,772 | 353,213 | 474,271 | 59,865 | 678,378 | 0.125 | 0.991 | 13 |
| ✓ | 1 | 10 | 0 | 38,068 | 107,347 | 133,019 | 207,736 | 410,955 | 550,753 | 66,882 | 746,211 | 0.625 | 1.067 | 7 |
| ✓ | 1 | 10 | 3 | 38,068 | 108,959 | 146,486 | 207,736 | 405,403 | 515,975 | 61,113 | 746,211 | 0.875 | 1.058 | 3 |

* **SR**: Success rate; **WLR**: Average wirelength ratio compared to PeF Li et al. (2022); **Rank**: **Rank** is determined primarily by SR, with a larger SR ensuring a better **Rank**. Within the same SR, a smaller WLR leads to a better **Rank**.

# Rand(n) represents selecting $n$ random instances from the retriever introduced in Sec. 4.1;

- `initAccpRate` $\in \{0.80, 0.85, 0.90\}$, which means the initial acceptance rate of the simulated annealing algorithm. High `initAccpRate` allows broad exploration and helps escape local optima but slows down convergence.

- `annealingRatioDecrease` $\in \{0.40, 0.60, 0.80\}$, which means the temperature decay rate of the simulated annealing algorithm. High decay rate (slow cooling) enhances thorough exploration and reduces the risk of premature convergence but extends computation time.

Note that there are a total of $8 \times (3 \times 4 \times 3 \times 2 + 4 \times 3 \times 3) = 864$ instances in the database.

## D.5 PARAMETERS OF BASELINES

For the default settings used in the baseline methods PeF (Li et al., 2022) and ECS (Chiou et al., 2016), we configure the hyperparameters as follows:

- For PeF (Li et al., 2022), we set `grid_size` $= 128$, `amplify_ratio` $= 1.15$, `init_lr` $= 0.1$, and `density_weight` $= 100000$.

- For ECS (Chiou et al., 2016), we set `amplifyRatio` $= 1.20$, `initAccpRate` $= 0.85$, and `annealingRatioDecrease` $= 0.60$.

Note that the amplify ratio in ECS is higher than that in PeF to avoid failure in finding non-overlapping solutions for certain circuits. Further details regarding these hyperparameters can be found in Appendix D.4.

# E  OTHER EXPERIMENTAL RESULTS

## E.1 DETAILS OF ABLATION STUDIES

We conduct the ablation study under the 4th requirement in Table 2 in Sec. 5.3, as it features a strict OR criterion, and employ GPT-4o-mini as the LLM backbone due to its lower cost. Additionally, we select the hyperparameters $k_1$ and $k_2$ from $k_1 \in \{1, 3\}$ and $k_2 \in \{5, 10\}$ in Sec. 4.1, and iters from iters $\in \{0, 3\}$ in Sec. 4.3. It is also crucial to evaluate the impact of integrating the model evaluation agent $h_\phi$. The best configuration is achieved with hyperparameters $k_1 = 3$, $k_2 = 10$, iters $= 3$, and with the inclusion of $h_\phi$.

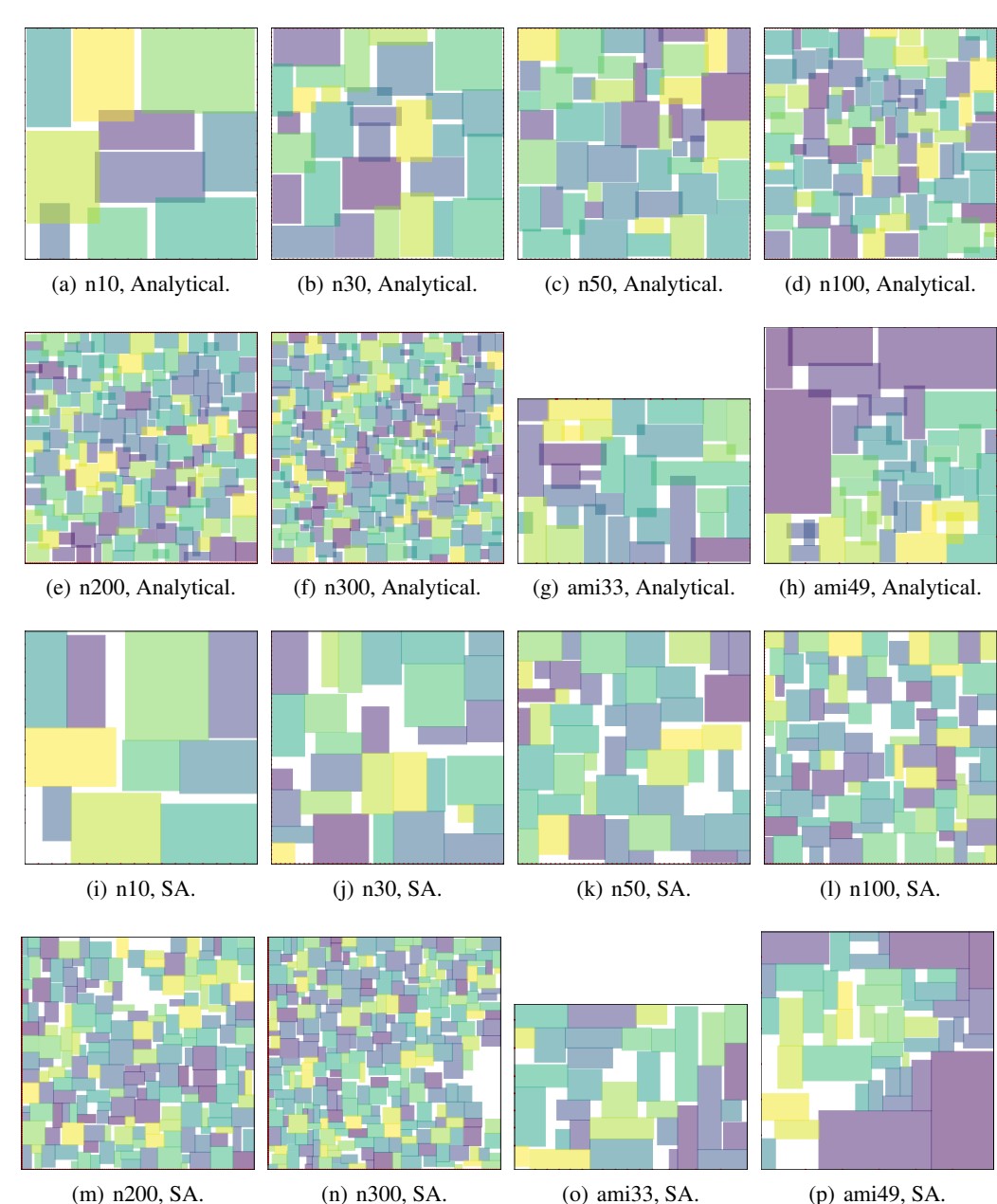

(a) n10, Analytical.  (b) n30, Analytical.  (c) n50, Analytical.  (d) n100, Analytical.

(e) n200, Analytical.  (f) n300, Analytical.  (g) ami33, Analytical.  (h) ami49, Analytical.

(i) n10, SA.  (j) n30, SA.  (k) n50, SA.  (l) n100, SA.

(m) n200, SA.  (n) n300, SA.  (o) ami33, SA.  (p) ami49, SA.

Figure 7: Visualization of LLM4Floorplan results. The first two rows represent layouts generated using an analytical floorplanning backbone, which are more compact but include some overlapping areas. The last two rows show layouts from a simulated annealing (SA)-based floorplanning backbone, which are more spaced out and ensure no overlaps.

## E.2 VISUALIZATION OF LLM4FLOORPLAN RESULTS

We present the visualization of LLM4Floorplan results in Fig. 7. The first two rows show results using an analytical floorplanning backbone, while the last two rows display results from a simulated annealing (SA)-based floorplanning backbone. Overall, the analytical approaches produce more compact layouts, though they retain some overlapping areas. In contrast, the SA-based approaches yield looser layouts, ensuring no overlapping areas. All visualizations effectively demonstrate the excellent performance of LLM4Floorplan.

Table 8: Floorplan results for requirement 1. Results unsatisfactory to meet OR $\leq 5\%$ criteria are in gray. The first rank is shown in cyan.

| Method | LLM Backbones | Metric | Circuits | | | | | | | | Overall Metrics | | |
|---|---|---|---|---|---|---|---|---|---|---|---|---|---|
| | | | n10 | n30 | n50 | n100 | n200 | n300 | ami33 | ami49 | SR*↑ | WLR*↓ | Rank*↓ |
| PeF (Li et al., 2022) | | HPWL | 37,097 | 104,488 | 130,589 | 198,685 | 361,313 | 480,571 | 59,061 | 725,235 | 1.000 | 1.000 | 6 |
| | | OR | 0.0148 | 0.0189 | 0.0207 | 0.0201 | 0.0271 | 0.0372 | 0.0219 | 0.0301 | | | |
| DRAT | DeepSeek-Chat | HPWL | 35,797 | 102,152 | 126,336 | 195,268 | 354,605 | 473,402 | 61,981 | 745,263 | 1.000 | 0.992 | 5 |
| | | OR | 0.0241 | 0.0231 | 0.0349 | 0.0303 | 0.0466 | 0.0624 | 0.0459 | 0.0334 | | | |
| | DeepSeek-Coder | HPWL | 39,201 | 102,650 | 133,187 | 195,713 | 379,148 | 481,584 | 61,120 | 737,008 | 1.000 | 1.018 | 8 |
| | | OR | 0.0089 | 0.0237 | 0.0125 | 0.0287 | 0.0169 | 0.0296 | 0.0204 | 0.0107 | | | |
| | GPT-3.5 | HPWL | 38,464 | 106,658 | 126,670 | 194,066 | 369,972 | 494,165 | 58,192 | 733,926 | 1.000 | 1.007 | 7 |
| | | OR | 0.0214 | 0.0082 | 0.0466 | 0.0353 | 0.0155 | 0.0259 | 0.0239 | 0.0256 | | | |
| | GPT-4 | HPWL | 34,967 | 99,652 | 124,546 | 189,354 | 346,186 | 456,867 | 60,194 | 708,398 | 0.625 | 0.956 | 10 |
| | | OR | 0.0398 | 0.0459 | 0.0263 | 0.0483 | 0.0545 | 0.0709 | 0.0500 | 0.0529 | | | |
| | GPT-4o-mini | HPWL | 35,896 | 99,730 | 126,379 | 194,375 | 367,964 | 474,257 | 57,673 | 710,099 | 1.000 | 0.979 | 3 |
| | | OR | 0.0215 | 0.0470 | 0.0281 | 0.0335 | 0.0264 | 0.0359 | 0.0288 | 0.0190 | | | |
| | GPT-4o | HPWL | 34,937 | 99,620 | 123,792 | 193,461 | 340,055 | 458,696 | 56,521 | 674,498 | 0.500 | 0.949 | 11 |
| | | OR | 0.0377 | 0.0514 | 0.0404 | 0.0452 | 0.0983 | 0.0903 | 0.0574 | 0.0413 | | | |
| | Claude-3.5 | HPWL | 35,479 | 99,777 | 125,941 | 190,482 | 348,836 | 464,595 | 56,330 | 669,894 | 0.750 | 0.956 | 9 |
| | | OR | 0.0307 | 0.0484 | 0.0347 | 0.0495 | 0.0584 | 0.0647 | 0.0476 | 0.0339 | | | |
| LLM4Floorplan | GPT-4o-mini | HPWL | 35,589 | 102,444 | 127,492 | 193,986 | 353,332 | 478,474 | 63,378 | 700,512 | 1.000 | 0.988 | 4 |
| | | OR | 0.0230 | 0.0234 | 0.0327 | 0.0291 | 0.0414 | 0.0430 | 0.0163 | 0.0257 | | | |
| | GPT-4o | HPWL | 34,907 | 99,704 | 124,116 | 196,169 | 353,526 | 469,933 | 58,663 | 700,512 | 1.000 | 0.969 | 2 |
| | | OR | 0.0387 | 0.0446 | 0.0397 | 0.0421 | 0.0459 | 0.0438 | 0.0475 | 0.0257 | | | |
| | Claude-3.5 | HPWL | 34,966 | 100,071 | 124,701 | 190,605 | 357,754 | 469,875 | 55,795 | 675,684 | 1.000 | 0.957 | 1 |
| | | OR | 0.0374 | 0.0429 | 0.0304 | 0.0426 | 0.0374 | 0.0462 | 0.0475 | 0.0420 | | | |

*  **SR**: Success rate; **WLR**: Average wirelength ratio compared to baseline PeF Li et al. (2022); **Rank**: Rank is determined primarily by SR, with a larger SR ensuring a better Rank. Within the same SR, a smaller WLR leads to a better Rank.

### E.3  Detailed Results in Sec. 5.2 and Sec. 5.3

We present the detailed results, including the Half-Perimeter Wire Length (HPWL), Overlapping Ratio (OR), Objective Value (OV), and adjacency metrics, in Table 8 to Table 13, which correspond to the comprehensive results shown in Table 3. It is important to note that Table 8 and Table 9 specifically align with the results shown in Table 1.

Table 9: Floorplan results for requirement 2. Failed results are displayed as 'N/A'. The first rank is shown in cyan.

| Method | LLM Backbones | Metric | Circuits | | | | | | | | Overall Metrics | | |
|---|---|---|---|---|---|---|---|---|---|---|---|---|---|
| | | | n10 | n30 | n50 | n100 | n200 | n300 | ami33 | ami49 | SR*↑ | WLR*↓ | Rank*↓ |
| ECS (Chiou et al., 2016) | | HPWL | 40,082 | 123,022 | 168,848 | 295,387 | 561,956 | 848,366 | 82,454 | 1,445,688 | 1.000 | 1.000 | 4 |
| | | OR | 0.0000 | 0.0000 | 0.0000 | 0.0000 | 0.0000 | 0.0000 | 0.0000 | 0.0000 | | | |
| DRAT | DeepSeek-Chat | HPWL | N/A | N/A | N/A | N/A | N/A | N/A | N/A | N/A | N/A | N/A | N/A |
| | | OR | N/A | N/A | N/A | N/A | N/A | N/A | N/A | N/A | | | |
| | DeepSeek-Coder | HPWL | N/A | N/A | N/A | N/A | N/A | N/A | N/A | N/A | N/A | N/A | N/A |
| | | OR | N/A | N/A | N/A | N/A | N/A | N/A | N/A | N/A | | | |
| | GPT-3.5 | HPWL | N/A | 123,102 | 172,105 | N/A | N/A | N/A | N/A | N/A | 0.250 | 1.010 | 5 |
| | | OR | N/A | 0.0000 | 0.0000 | N/A | N/A | N/A | N/A | N/A | | | |
| | GPT-4 | HPWL | N/A | N/A | N/A | N/A | N/A | N/A | N/A | N/A | N/A | N/A | N/A |
| | | OR | N/A | N/A | N/A | N/A | N/A | N/A | N/A | N/A | | | |
| | GPT-4o-mini | HPWL | N/A | N/A | N/A | N/A | N/A | N/A | 76,104 | N/A | 0.125 | 0.982 | 6 |
| | | OR | N/A | N/A | N/A | N/A | N/A | N/A | 0.0000 | N/A | | | |
| | GPT-4o | HPWL | N/A | N/A | N/A | N/A | N/A | 839,226 | N/A | N/A | 0.125 | 0.989 | 7 |
| | | OR | N/A | N/A | N/A | N/A | N/A | 0.0000 | N/A | N/A | | | |
| | Claude-3.5 | HHPWL | N/A | N/A | N/A | N/A | N/A | N/A | N/A | N/A | N/A | N/A | N/A |
| | | OR | N/A | N/A | N/A | N/A | N/A | N/A | N/A | N/A | | | |
| LLM4Floorplan | GPT-4o-mini | HPWL | 39,130 | 126,168 | 168,198 | 283,538 | 568,540 | 863,628 | 70,769 | 1,442,816 | 1.000 | 0.981 | 3 |
| | | OR | 0.0000 | 0.0000 | 0.0000 | 0.0000 | 0.0000 | 0.0000 | 0.0000 | 0.0000 | | | |
| | GPT-4o | HPWL | 35,660 | 120,142 | 164,987 | 280,063 | 553,580 | 829,065 | 79,417 | 1,462,553 | 1.000 | 0.966 | 1 |
| | | OR | 0.0000 | 0.0000 | 0.0000 | 0.0000 | 0.0000 | 0.0000 | 0.0000 | 0.0000 | | | |
| | Claude-3.5 | HPWL | 37,086 | 118,800 | 166,750 | 284,426 | 551,714 | 832,472 | 83,730 | 1,400,434 | 1.000 | 0.974 | 2 |
| | | OR | 0.0000 | 0.0000 | 0.0000 | 0.0000 | 0.0000 | 0.0000 | 0.0000 | 0.0000 | | | |

*  **SR**: Success rate; **WLR**: Average wirelength ratio compared to baseline ECS (Chiou et al., 2016); **Rank**: Rank is determined primarily by SR, with a larger SR ensuring a better Rank. Within the same SR, a smaller WLR leads to a better Rank.

Table 10: Floorplan results for requirement 3. Results unsatisfactory to meet OR $\leq 2.5\%$ criteria are in gray. The first rank is shown in cyan.

| Method | LLM Backbones | Metric | Circuits | | | | | | | | Overall Metrics | | |
|---|---|---|---|---|---|---|---|---|---|---|---|---|---|
| | | | n10 | n30 | n50 | n100 | n200 | n300 | ami33 | ami49 | SR*↑ | WLR*↓ | Rank*↓ |
| PeF (Li et al., 2022) | | HPWL | 37,097 | 104,488 | 130,589 | 198,685 | 361,313 | 481,350 | 59,061 | 740,577 | 0.750 | 1.000 | 7 |
| | | OR | 0.0148 | 0.0189 | 0.0207 | 0.0201 | 0.0271 | 0.0340 | 0.0219 | 0.244 | | | |
| ECS (Chiou et al., 2016) | | HPWL | 40,082 | 123,022 | 168,848 | 295,387 | 561,956 | 848,366 | 82,454 | 1,445,688 | 1.000 | 1.463 | 3 |
| | | OR | 0.0000 | 0.0000 | 0.0000 | 0.0000 | 0.0000 | 0.0000 | 0.0000 | 0.0000 | | | |
| DRAT | DeepSeek-Chat | HPWL | 36,021 | 102,791 | 130,589 | 198,207 | 361,313 | 479,061 | 60,468 | 740,577 | 0.625 | 0.996 | 10 |
| | | OR | 0.0231 | 0.0236 | 0.0207 | 0.0213 | 0.0271 | 0.0344 | 0.0372 | 0.0244 | | | |
| | DeepSeek-Coder | HPWL | 35,957 | 104,215 | 128,182 | 195,891 | 353,213 | 479,339 | 61,120 | 712,357 | 0.875 | 0.988 | 5 |
| | | OR | 0.0237 | 0.0203 | 0.0208 | 0.0216 | 0.0218 | 0.0252 | 0.0164 | 0.022 | | | |
| | GPT-3.5 | HPWL | 38,464 | 106,279 | 133,226 | 195,363 | 347,141 | 479,339 | 61,495 | 734,357 | 0.750 | 1.006 | 8 |
| | | OR | 0.0214 | 0.0104 | 0.021 | 0.0245 | 0.0848 | 0.0252 | 0.0166 | 0.0114 | | | |
| | GPT-4 | HPWL | 37,767 | 102,212 | 131,390 | 191,094 | 346,257 | 476,975 | 67,689 | 724,870 | 0.125 | 0.985 | 12 |
| | | OR | 0.0518 | 0.0204 | 0.0279 | 0.0359 | 0.0354 | 0.0260 | 0.0347 | 0.0428 | | | |
| | GPT-4o-mini | HPWL | 36,745 | 105,207 | 127,756 | 195,363 | 353,213 | 479,461 | 61,120 | 712,357 | 0.875 | 0.991 | 4 |
| | | OR | 0.0281 | 0.0175 | 0.0163 | 0.0245 | 0.0218 | 0.0229 | 0.0164 | 0.022 | | | |
| | GPT-4o | HPWL | 34,969 | 100,802 | 129,102 | 199,875 | 354,541 | 469,863 | 58,403 | 689,266 | 0.375 | 0.970 | 11 |
| | | OR | 0.0374 | 0.0260 | 0.0181 | 0.0098 | 0.0164 | 0.0427 | 0.0519 | 0.0289 | | | |
| | Claude-3.5 | HPWL | 37,364 | 107,551 | 133,050 | 203,918 | 371,090 | 493,201 | 62,428 | 736,063 | 1.000 | 1.023 | 2 |
| | | OR | 0.0123 | 0.0059 | 0.0113 | 0.0125 | 0.0126 | 0.0136 | 0.0104 | 0.0104 | | | |
| LLM4Floorplan | GPT-4o-mini | HPWL | 39,986 | 104,488 | 128,327 | 198,823 | 355,149 | 476,975 | 67,676 | 699,066 | 0.750 | 1.016 | 9 |
| | | OR | 0.0228 | 0.0189 | 0.016 | 0.0202 | 0.0223 | 0.0260 | 0.031 | 0.0248 | | | |
| | GPT-4o | HPWL | 42,025 | 102,155 | 128,327 | 194,908 | 355,149 | 476,975 | 63,378 | 706,040 | 0.875 | 1.009 | 6 |
| | | OR | 0.0246 | 0.0216 | 0.016 | 0.0228 | 0.0223 | 0.0260 | 0.0163 | 0.0247 | | | |
| | Claude-3.5 | HPWL | 39,358 | 103,640 | 128,205 | 196,132 | 351,782 | 477,893 | 61,808 | 708,719 | 1.000 | 0.999 | 1 |
| | | OR | 0.0105 | 0.0250 | 0.0241 | 0.0215 | 0.0246 | 0.0210 | 0.0196 | 0.0163 | | | |

* **SR**: Success rate; **WLR**: Average wirelength ratio compared to baseline PeF Li et al. (2022); **Rank**: Rank is determined primarily by SR, with a larger SR ensuring a better Rank. Within the same SR, a smaller WLR leads to a better Rank.

Table 11: Floorplan results for requirement 4. Results unsatisfactory to meet OR $\leq 1.0\%$ criteria are in gray while the failed results are displayed as 'N/A'. The first rank is shown in cyan.

| Method | LLM Backbones | Metric | Circuits | | | | | | | | Overall Metrics | | |
|---|---|---|---|---|---|---|---|---|---|---|---|---|---|
| | | | n10 | n30 | n50 | n100 | n200 | n300 | ami33 | ami49 | SR*↑ | WLR*↓ | Rank*↓ |
| PeF (Li et al., 2022) | | HPWL | 37,128 | 104,564 | 130,944 | 198,685 | 361,313 | 481,350 | 59,061 | 740,577 | 0.000 | 1.000 | 11 |
| | | OR | 0.0136 | 0.0177 | 0.0192 | 0.0201 | 0.0271 | 0.0340 | 0.0219 | 0.0244 | | | |
| ECS (Chiou et al., 2016) | | HPWL | 40,082 | 123,022 | 168,848 | 295,387 | 561,956 | 848,366 | 82,454 | 1,445,688 | 1.000 | 1.462 | 3 |
| | | OR | 0.0000 | 0.0000 | 0.0000 | 0.0000 | 0.0000 | 0.0000 | 0.0000 | 0.0000 | | | |
| DRAT | DeepSeek-Chat | HPWL | 40,423 | 110,524 | 134,501 | 204,143 | 369,036 | 497,453 | 63,852 | 753,063 | 0.625 | 1.044 | 6 |
| | | OR | 0.0061 | 0.0108 | 0.0094 | 0.0084 | 0.0098 | 0.0101 | 0.0153 | 0.0096 | | | |
| | DeepSeek-Coder | HPWL | 40,423 | 110,524 | 134,501 | 204,143 | 369,036 | 497,453 | 63,852 | 753,063 | 0.625 | 1.044 | 6 |
| | | OR | 0.0061 | 0.0108 | 0.0094 | 0.0084 | 0.0098 | 0.0101 | 0.0153 | 0.0096 | | | |
| | GPT-3.5 | HPWL | 39,920 | 107,934 | 145,215 | 208,439 | 564,052 | 841,553 | 90,602 | 841,713 | 1.000 | 1.281 | 2 |
| | | OR | 0.0000 | 0.0064 | 0.0000 | 0.0087 | 0.0000 | 0.0000 | 0.0000 | 0.0096 | | | |
| | GPT-4 | HPWL | 40,423 | 101,096 | N/A | 195,177 | 356,504 | 497,818 | 58,277 | 833,810 | 0.250 | 1.025 | 10 |
| | | OR | 0.0061 | 0.0403 | N/A | 0.0344 | 0.0259 | 0.0095 | 0.0417 | 0.0169 | | | |
| | GPT-4o-mini | HPWL | 38,593 | 105,016 | 134,501 | 199,757 | 370,255 | 495,216 | 60,223 | 750,760 | 0.250 | 1.020 | 9 |
| | | OR | 0.0156 | 0.0191 | 0.0094 | 0.0215 | 0.0088 | 0.0120 | 0.0151 | 0.0168 | | | |
| | GPT-4o | HPWL | 44,301 | 101,508 | 124,657 | 199,494 | 381,495 | 469,863 | 63,613 | 753,595 | 0.000 | 1.031 | 12 |
| | | OR | 0.0551 | 0.0397 | 0.0217 | 0.0229 | 0.0244 | 0.0427 | 0.0656 | 0.0295 | | | |
| | Claude-3.5 | HPWL | 37,741 | 108,169 | 133,744 | 203,918 | 369,036 | 497,453 | 67,386 | 753,063 | 0.500 | 1.039 | 8 |
| | | OR | 0.0119 | 0.0072 | 0.0077 | 0.0125 | 0.0098 | 0.0101 | 0.0185 | 0.0096 | | | |
| LLM4Floorplan | GPT-4o-mini | HPWL | 37,915 | 110,982 | 138,298 | 208,069 | 385,422 | 514,722 | 63,745 | 787,709 | 1.000 | 1.058 | 1 |
| | | OR | 0.0087 | 0.0079 | 0.0055 | 0.0077 | 0.0076 | 0.0076 | 0.0055 | 0.0086 | | | |
| | GPT-4o | HPWL | 37,960 | 106,801 | 138,362 | 211,788 | 573,158 | 502,790 | 58,743 | 736,979 | 0.750 | 1.098 | 5 |
| | | OR | 0.0093 | 0.0086 | 0.0058 | 0.0080 | 0.0000 | 0.0100 | 0.0117 | 0.0148 | | | |
| | Claude-3.5 | HPWL | 40,423 | 111,023 | 133,776 | 203,530 | 368,428 | 507,746 | 69,019 | 738,207 | 0.875 | 1.055 | 4 |
| | | OR | 0.0061 | 0.0066 | 0.0096 | 0.0100 | 0.0093 | 0.0059 | 0.0109 | 0.0085 | | | |

* **SR**: Success rate; **WLR**: Average wirelength ratio compared to baseline PeF Li et al. (2022); **Rank**: Rank is determined primarily by SR, with a larger SR ensuring a better Rank. Within the same SR, a smaller WLR leads to a better Rank.

Table 12: Floorplan results for requirement 5, which minimize objective value (OV) = WL× (1+10×OR). The first rank is shown in cyan.

| Method | LLM Backbones | Metric | Circuits | | | | | | | | Overall Metrics | |
|---|---|---|---|---|---|---|---|---|---|---|---|---|
| | | | n10 | n30 | n50 | n100 | n200 | n300 | ami33 | ami49 | Obj. VR*↓ | Rank↓ |
| PeF (Li et al., 2022) | | HPWL | 37,128 | 104,564 | 130,944 | 198,685 | 361,313 | 481,350 | 59,061 | 740,577 | | |
| | | OR | 0.0136 | 0.0177 | 0.0192 | 0.0201 | 0.0271 | 0.0340 | 0.0219 | 0.0244 | 1.000 | 10 |
| | | OV | 42,177 | 123,072 | 156,085 | 238,621 | 459,229 | 645,009 | 71,995 | 921,278 | | |
| ECS (Chiou et al., 2016) | | HPWL | 40,082 | 123,022 | 168,848 | 295,387 | 561,956 | 848,366 | 82,454 | 1,445,688 | | |
| | | OR | 0.0000 | 0.0000 | 0.0000 | 0.0000 | 0.0000 | 0.0000 | 0.0000 | 0.0000 | 1.190 | 12 |
| | | OV | 40,082 | 123,022 | 168,848 | 295,387 | 561,956 | 848,366 | 82,454 | 1,445,688 | | |
| DRAT | DeepSeek-Chat | HPWL | 39,082 | 108,292 | 133,841 | 203,873 | 371,209 | 494,489 | 59,061 | 740,577 | | |
| | | OR | 0.0062 | 0.0097 | 0.0154 | 0.0160 | 0.0218 | 0.0344 | 0.0219 | 0.0244 | 0.993 | 9 |
| | | OV | 41,505 | 118,796 | 154,453 | 236,493 | 452,133 | 664,593 | 71,995 | 921,278 | | |
| | DeepSeek-Coder | HPWL | 37,128 | 104,564 | 135,186 | 198,685 | 361,313 | 497,453 | 59,061 | 774,645 | | |
| | | OR | 0.0136 | 0.0177 | 0.0084 | 0.0201 | 0.0271 | 0.0101 | 0.0219 | 0.0093 | 0.963 | 7 |
| | | OV | 42,177 | 123,072 | 146,542 | 238,621 | 459,229 | 547,696 | 71,995 | 846,687 | | |
| | GPT-3.5 | HPWL | 40,052 | 123,236 | 145,215 | 204,143 | 568,921 | 841,553 | 63,852 | 1,530,540 | | |
| | | OR | 0.0000 | 0.0000 | 0.0000 | 0.0084 | 0.0000 | 0.0000 | 0.0153 | 0.0000 | 1.130 | 11 |
| | | OV | 40,052 | 123,236 | 145,215 | 221,291 | 568,921 | 841,553 | 73,621 | 1,530,540 | | |
| | GPT-4 | HPWL | 38,558 | 106,682 | 136,276 | 204,143 | 370,222 | 495,926 | 64,145 | 739,561 | | |
| | | OR | 0.0093 | 0.0076 | 0.0064 | 0.0084 | 0.0099 | 0.0141 | 0.0097 | 0.0065 | 0.923 | 2 |
| | | OV | 42,144 | 114,790 | 144,998 | 221,291 | 406,874 | 565,852 | 70,367 | 787,632 | | |
| | GPT-4o-mini | HPWL | 37,915 | 106,369 | 134,501 | 204,384 | 368,428 | 495,216 | 60,223 | 750,794 | | |
| | | OR | 0.0087 | 0.0109 | 0.0094 | 0.0090 | 0.0093 | 0.0120 | 0.0151 | 0.0168 | 0.933 | 4 |
| | | OV | 41,214 | 117,963 | 147,144 | 222,779 | 402,692 | 554,642 | 69,317 | 876,927 | | |
| | GPT-4o | HPWL | 37,071 | 107,637 | 129,102 | 207,370 | 371,722 | 477,338 | 63,512 | 700,512 | | |
| | | OR | 0.0452 | 0.0071 | 0.0181 | 0.0087 | 0.0147 | 0.0212 | 0.0081 | 0.0257 | 0.984 | 8 |
| | | OV | 53,827 | 115,279 | 152,469 | 225,411 | 426,365 | 578,534 | 68,656 | 880,544 | | |
| | Claude-3.5 | HPWL | 37,364 | 107,551 | 133,798 | 204,384 | 370,255 | 493,201 | 67,386 | 736,063 | | |
| | | OR | 0.0123 | 0.0059 | 0.0102 | 0.0090 | 0.0088 | 0.0136 | 0.0185 | 0.0104 | 0.942 | 5 |
| | | OV | 41,960 | 113,897 | 147,445 | 222,779 | 402,837 | 560,276 | 79,852 | 812,614 | | |
| LLM4Floorplan | GPT-4o-mini | HPWL | 39,789 | 110,637 | 138,726 | 211,690 | 412,302 | 521,808 | 63,378 | 735,049 | | |
| | | OR | 0.0010 | 0.0048 | 0.0039 | 0.0066 | 0.0031 | 0.0049 | 0.0163 | 0.0060 | 0.926 | 3 |
| | | OV | 40,187 | 115,948 | 144,136 | 225,662 | 425,083 | 547,377 | 73,709 | 779,152 | | |
| | GPT-4o | HPWL | 38,906 | 118,714 | 130,589 | 203,757 | 378,257 | 501,593 | 59,584 | 745,384 | | |
| | | OR | 0.0032 | 0.0000 | 0.0207 | 0.0146 | 0.0104 | 0.0170 | 0.0156 | 0.0157 | 0.952 | 6 |
| | | OV | 40,151 | 118,714 | 157,621 | 233,506 | 417,596 | 586,864 | 68,879 | 862,409 | | |
| | Claude-3.5 | HPWL | 37,642 | 107,491 | 135,045 | 206,712 | 370,255 | 501,912 | 58,733 | 737,309 | | |
| | | OR | 0.0000 | 0.0061 | 0.0113 | 0.0061 | 0.0088 | 0.0088 | 0.0158 | 0.0089 | 0.905 | 1 |
| | | OV | 37,642 | 114,048 | 150,305 | 219,321 | 402,837 | 546,080 | 68,013 | 802,930 | | |

* **Obj. VR**: Objective value ratio is compared with PeF Li et al. (2022). The objective value is defined as $\sum_{e_i \in \mathcal{E}} \text{HPWL}(e_i) \times (1 + 10 \times \text{OR})$, as described in Table 2; **Rank**: A smaller Obj. VR leads to a better Rank.

Table 13: Floorplan results for requirement 6. An additional constraint is introduced, requiring the first two blocks to be adjacent. The top-ranked result is highlighted in cyan.

| Method | LLM Backbones | Metric | Circuits | | | | | | | | Overall Metrics | | |
|---|---|---|---|---|---|---|---|---|---|---|---|---|---|
| | | | n10 | n30 | n50 | n100 | n200 | n300 | ami33 | ami49 | SR*↑ | WLR*↓ | Rank*↓ |
| PeF (Li et al., 2022) | | HPWL | 37,097 | 104,488 | 130,589 | 198,685 | 361,313 | 480,571 | 59,061 | 725,235 | 0.000 | 1.000 | 10 |
| | | OR | 0.0245 | 0.0260 | 0.0148 | 0.0203 | 0.0307 | 0.0327 | 0.0220 | 0.0119 | | | |
| | | Adj. | ✗ | ✗ | ✗ | ✗ | ✗ | ✗ | ✗ | ✗ | | | |
| ECS (Chiou et al., 2016) | | HPWL | 40,082 | 123,022 | 168,848 | 295,387 | 561,956 | 848,366 | 82,454 | 1,445,688 | 0.000 | 1.469 | 12 |
| | | OR | 0.0000 | 0.0000 | 0.0000 | 0.0000 | 0.0000 | 0.0000 | 0.0000 | 0.0000 | | | |
| | | Adj. | ✗ | ✗ | ✗ | ✗ | ✗ | ✗ | ✗ | ✗ | | | |
| DRAT | DeepSeek-Chat | HPWL | 37,755 | 104,473 | 128,073 | 195,713 | 361,548 | 479,061 | 59,821 | 726,519 | 0.000 | 0.999 | 9 |
| | | OR | 0.0118 | 0.0129 | 0.0248 | 0.0287 | 0.0212 | 0.0344 | 0.0300 | 0.0127 | | | |
| | | Adj. | ✗ | ✗ | ✗ | ✗ | ✗ | ✗ | ✗ | ✗ | | | |
| | DeepSeek-Coder | HPWL | 35,957 | 102,601 | 128,182 | 191,094 | 345,515 | 462,491 | 58,277 | 712,357 | 0.000 | 0.973 | 8 |
| | | OR | 0.0237 | 0.0304 | 0.0208 | 0.0359 | 0.0366 | 0.0398 | 0.0417 | 0.0220 | | | |
| | | Adj. | ✗ | ✗ | ✗ | ✗ | ✗ | ✗ | ✗ | ✗ | | | |
| | GPT-3.5 | HPWL | 37,734 | 106,355 | 129,011 | 198,685 | 362,416 | 471,728 | 62,677 | 757,385 | 0.000 | 1.014 | 11 |
| | | OR | 0.0113 | 0.0230 | 0.0248 | 0.0201 | 0.0320 | 0.0396 | 0.0223 | 0.0146 | | | |
| | | Adj. | ✗ | ✗ | ✗ | ✗ | ✗ | ✗ | ✗ | ✗ | | | |
| | GPT-4 | HPWL | 34,058 | 102,261 | 121,931 | 186,389 | 353,213 | 460,442 | 57,011 | 684,310 | 0.125 | 0.952 | 5 |
| | | OR | 0.0624 | 0.0565 | 0.0431 | 0.0605 | 0.0218 | 0.0633 | 0.0514 | 0.0453 | | | |
| | | Adj. | ✗ | ✓ | ✗ | ✗ | ✗ | ✗ | ✗ | ✗ | | | |
| | GPT-4o-mini | HPWL | 38,387 | 102,866 | 127,101 | 193,870 | 353,640 | 468,824 | 58,798 | 718,985 | 0.250 | 0.989 | 4 |
| | | OR | 0.0164 | 0.0306 | 0.0300 | 0.0324 | 0.044 | 0.0506 | 0.0352 | 0.0200 | | | |
| | | Adj. | ✓ | ✗ | ✓ | ✗ | ✗ | ✗ | ✗ | ✗ | | | |
| | GPT-4o | HPWL | 35,102 | 102,246 | 124,393 | 190,737 | 345,286 | 463,231 | 58,876 | 721,716 | 0.125 | 0.969 | 6 |
| | | OR | 0.0366 | 0.0189 | 0.0341 | 0.0299 | 0.0337 | 0.0480 | 0.065 | 0.0641 | | | |
| | | Adj. | ✗ | ✗ | ✓ | ✗ | ✗ | ✗ | ✗ | ✗ | | | |
| | Claude-3.5 | HPWL | 39,118 | 110,139 | 137,632 | 205,483 | 372,387 | 495,796 | 67,280 | 741,721 | 1.000 | 1.053 | 1 |
| | | OR | 0.0168 | 0.0135 | 0.0154 | 0.0164 | 0.0160 | 0.0128 | 0.0325 | 0.0259 | | | |
| | | Adj. | ✓ | ✓ | ✓ | ✓ | ✓ | ✓ | ✓ | ✓ | | | |
| LLM4Floorplan | GPT-4o-mini | HPWL | 35,916 | 102,444 | 124,678 | 193,991 | 348,012 | 467,345 | 55,311 | 1,836,250 | 0.125 | 1.161 | 7 |
| | | OR | 0.0214 | 0.0234 | 0.0523 | 0.0450 | 0.0676 | 0.064 | 0.0462 | 0.0442 | | | |
| | | Adj. | ✗ | ✗ | ✗ | ✗ | ✗ | ✗ | ✗ | ✗ | | | |
| | GPT-4o | HPWL | 36,542 | 101,145 | 127,753 | 192,034 | 343,713 | 459,934 | 61,111 | 700,512 | 0.375 | 0.976 | 3 |
| | | OR | 0.0032 | 0.0000 | 0.0207 | 0.0146 | 0.0104 | 0.0170 | 0.0156 | 0.0157 | | | |
| | | Adj. | ✗ | ✗ | ✓ | ✗ | ✗ | ✓ | ✗ | ✓ | | | |
| | Claude-3.5 | HPWL | 35,618 | 101,155 | 125,924 | 198,240 | 348,522 | 464,918 | 62,609 | 714,832 | 0.875 | 0.984 | 2 |
| | | OR | 0.0271 | 0.0360 | 0.0561 | 0.0223 | 0.0317 | 0.0399 | 0.0630 | 0.0473 | | | |
| | | Adj. | ✗ | ✓ | ✓ | ✓ | ✓ | ✓ | ✓ | ✓ | | | |

[*] **SR**: Success Rate. A design simultaneously meet the overlapping requirement and the adjacency requirement is regarded as successful. **WLR**: Average wirelength ratio compared to baseline PeF (Li et al., 2022); **Rank**: Rank is determined primarily by SR, with a larger SR ensuring a better Rank. Within the same SR, a smaller WLR leads to a better Rank.

