# OpenReview forum: "Towards LLM4Floorplan: Agents Can Do What Engineers Do in Chip Design"
_ICLR.cc/2025/Conference — ICLR 2025 Conference Withdrawn Submission_

### Official Review · Reviewer_tZbE · 2024-10-17

**Soundness:** 2
**Presentation:** 2
**Contribution:** 3
**Rating:** 5
**Confidence:** 4

**Summary:**

This paper proposes LLM4floorplan, a multi-LLM-agent framework for floorplan design. A search-cluster-based retriever is used to facilitate the LLM design. In addition, a Dynamic Retrieval-Augmented Thought (DRAT) prompting technique is employed to decompose the task into smaller tasks for better design. Experimental studies are performed to demonstrate the effectiveness of LLM4floorplan.

**Strengths:**

This idea of using LLMs for floorplan design is interesting and worth exploring.

**Weaknesses:**

1. The workflow of LLM4floorplan is not clear. Sections 4 and 5 could be further improved by making the terminologies and notations consistent.

2. From Tables 1 and 3, even without a retriever, DRAT with some LLMs seems to perform relatively well (sometimes even better than LLM4floorplan). This undermines Contribution 2, which I believe is the main contribution of this work.

3. There is no code available for reproduction.

4. References for AI-based chip design, such as [1,2], are missing.

[1] Mirhoseini, A., Goldie, A., Yazgan, M., Jiang, J., Songhori, E., Wang, S., Lee, Y.J., Johnson, E., Pathak, O., Bae, S., and Nazi, A., 2020. Chip placement with deep reinforcement learning. arXiv preprint arXiv:2004.10746.

[2] Goldie, A., Mirhoseini, A., Yazgan, M., Jiang, J.W., Songhori, E., Wang, S., Lee, Y.J., Johnson, E., Pathak, O., Nova, A., Pak, J. Addendum: A graph placement methodology for fast chip design. Nature. 2024 Sep 26:1-2.

**Questions:**

1. How are executed instances $\mathcal{S}$, circuit information $\mathcal{B}$, and the set of model selections $\mathcal{C}$ stored? The description in Section 4.1 is too vague.

2. The authors claim that LLM4Floorplan incorporates iterations as shown in Algorithm 1; however, it is not clear how the LLM fine-tunes previous results. It appears that the database is updated each iteration, but $r_1$ is not updated along with the iteration. It would make more sense if $r_1$ were also updated for each iteration in terms of algorithms.

3. Why is $k_2$ larger than $k_1$ in the experiment? Based on the discussions in Section 4.1, $k_2$ should be smaller than $k_1$. Am I missing something?

4. How is the database initialized?

5. What is the difference between the lower part and upper part of Table 1 for DRAT and LLM4Floorplan? Are PeF and ECS affecting the LLM4Floorplan? This is not clearly explained in the experimental section.

6. The discussion from lines 393 to 396, "This phenomenon implies that DRAT with powerful LLMs, e.g., GPT-4, GPT-4o, and Claude-3.5, might be even worse than other moderate LLMs as these powerful LLMs integrate too much case-specific guidance but ignore the domain-specific expertise," needs more evidence, such as more detailed demonstration examples. Normally, a larger model should have better performance.

7. How does LLM4Floorplan compare to [1,2]? Some evaluation metrics presented in [1,2] are interesting to explore, such as timing (e.g., Worst Negative Slack (WNS) and Total Negative Slack (TNS)), power, and area, etc.

[1] Mirhoseini, A., Goldie, A., Yazgan, M., Jiang, J., Songhori, E., Wang, S., Lee, Y.J., Johnson, E., Pathak, O., Bae, S., and Nazi, A., 2020. Chip placement with deep reinforcement learning. arXiv preprint arXiv:2004.10746.

[2] Goldie, A., Mirhoseini, A., Yazgan, M., Jiang, J.W., Songhori, E., Wang, S., Lee, Y.J., Johnson, E., Pathak, O., Nova, A., Pak, J. Addendum: A graph placement methodology for fast chip design. Nature. 2024 Sep 26:1-2.

8. The prompt in Section C seems incomplete for the ASSISTANT block. More discussions on how to use these prompts are needed.

---

### Official Review · Reviewer_gWF3 · 2024-10-26

**Soundness:** 3
**Presentation:** 3
**Contribution:** 2
**Rating:** 5
**Confidence:** 3

**Summary:**

This paper is laser focused on a physical design problem which is NP hard, and proposed an LLM4Floorplan agent which retrieves from external knowledge to solve the problem. Coupled with a novel DRAT prompting technique, the paper claims a significant boost of success rate from 0.250 to 0.875.

**Strengths:**

The authors conducted extensive experimental study involving multiple LLMs and circuits of different scales. The appendix provided useful illustrations and prompt examples to explain the problem.

**Weaknesses:**

For readers outside EDA or physical design space, it is hard to attribute contributions. The paper cited related works focusing on code generation (for design languages like Verilog, RTL), which is a well-recognized area with criteria, benchmarks, etc.

But this work looks to me to be an application of function calling and retrieval with no prior SOTA. I do not see much contribution to the LLM field, but instead a groundbreaking change to the physical design space probably deserving a dedicated forum therein.

**Questions:**

Given the same setting (prior instances in the knowedge base), what is the human (professional) performance and hourly cost to solve the same problem?

What is the performance of a NP-approximation algorithm to this problem?

Since GPT-4o (maybe others too) supports multi modality, can you provide the circuit picture and prompt it to directly solve the problem without seeking external knowledge? This will give a baseline to separate model knowledge and external knowledge.

---

### Official Review · Reviewer_RMpB · 2024-11-03

**Soundness:** 2
**Presentation:** 2
**Contribution:** 2
**Rating:** 3
**Confidence:** 3

**Summary:**

The paper proposes LLM4Floorplan for generating floorplans based on a requirement document and a given database of plans. The system retrieves an instance which includes a requirement document and a circuit, based on this a model is generated, based on the model parameters are retrieved, based on the model and the parameters code is retrieved, then the code is executed and evaluated, based on this comments are generated. The process is repeated a number of times.

**Strengths:**

* Improving floorplanning is an important problem with significant economical impact.
* The approach outperforms the other approaches on standard benchmarks so it seems to improve the state-of-the-art performance (however, I am not an expert in floorplanning so I cannot judge if PeF and ECS are the state-of-the-art solvers).

**Weaknesses:**

* I find it very hard to understand what is really done. At a high level it seems to make sense, but when I try to understand the details it gets confusing. The appendix helps a lot, but it is still not enough. Please see the question section.
* It is not clear what problem is really solved by the system.
* It is not clear why the proposal is framed as a multi-agent approach.
* It is not clear how well the approach would generalize to other problems.
* It is not clear why it would be a good idea to simulate how humans approach the problem.
* Looking at other papers about floorplanning it seems that it is common to also evaluate on benchmarks with obstacles. No such benchmarks are included (however, in the formalization of the problem in the appendix pre-placed blocks are included but it is not clear what type of constraints are included in the actual benchmark).

**Questions:**

Why would it be a good approach to simulate the workflow of humans, especailly since the problem is a well-defined NP-hard algorithmic problem? Why would it be easier for humans to specify the requirements in text rather than in something more formal? Since this is a common problem aren't there more specialized languages for specifying requirements? How would a human evaluate whether the generated floorplans actually correspond to the desired requiements? How much does requirements change between designs?

Why is it acceptable to have 5% overlap? Is this the maximum the legalization process can manage?

You state that "LLM4Floorplan faces challenges in handling highly novel designs due to its reliance on a predefined design database", how would you populate the predefined design database? How dependent is the final result on the quality of the database? How would you evaluate this?

How is the model C represented?

Is it correct that you generate code either for PeF (ANALYTICAL) or ECA (SA) so these are the solvers that are used to generate solutions?

Why do you represent the circuits as images?

How do you verify that the generated floorplans are correct?

How do you verify that the comments generated are correct?

 Since you get better results compared to other approaches, does this mean that you find variations of the designs in your database, i.e. do a local search around the designs in the database?

Why do you model the problem as four agents rather than as four steps in a sequential process? What is done in parallel? What properties or characteristics of agent-hood does your approach leverage?

---

### Official Review · Reviewer_CNXo · 2024-11-04

**Soundness:** 2
**Presentation:** 1
**Contribution:** 3
**Rating:** 3
**Confidence:** 3

**Summary:**

This paper presents an agent-based approach for using an LLM to solve the circuit board layout design problem known as floorplanning: how to arrange grid-aligned rectangular blocks into a fixed rectangular footprint subject to optimization goals (e.g. minimizing overlap) and constraints (fixed-position or fixed-size blocks). The approach builds on Retrieval-Augmented Thought, a prompting approach that applies RAG queries between chain-of-thought steps, by the inclusion of algorithmic _models_ that are executed between steps, amounting to a series of experimental solutions, which become retrievable data in future steps. The paper calls this approach *Dynamic* Retrieval-Augmented Though (DRAT), and claims that it is emulating human design processes.

The paper makes 2 primary contributions:

- The DRAT querying process, described above
- Search-Cluster-Based Retrieval: A querying strategy designed to find relevant but also diverse examples by first querying for relevant examples as K-nearest neighbors (KNN) under cosine similarity, then refining this for diversity by choosing cluster representatives under spectral clustering of the initial search results.

The authors show that under an objective-balancing ranking metric, the DRAT approach outperforms a PeF baseline approach.

**Strengths:**

This work is tackling a difficult and important problem; generative design in a sparse domain, and according to its evaluation metrics appears to perform quite well. It does so by making two technical contributions; a retrieval strategy that could be useful in a variety of RAG contexts, and well as a prompting strategy that attempts to emulate the iterative pattern of design processes.

The exposition of the search-cluster-based retriever as a general method is also quite clear.

**Weaknesses:**

The primary weakness of this paper lies in its exposition. The text of the paper contains insufficient detail to reproduce the method, and the details that are given are presented in an order that is difficult to follow and requires frequent reference to the Appendices to fill in critical details. I will give one detailed example to illustrate:

L194-195: The representation formats for S, B, and C are never specified. Knowing the representation of S and B is required to understand what the cosine distance used in the Relevance query (L208) is measuring, and similarly the spectral clustering. Furthermore, this is the first reference to the idea of a "model selection." The existence of executable code "models" has not been introduced yet; the first place in the paper that their existence could be deduced is reading Algorithm 1 in detail, which is first referenced a page later (L263). "Model selection" is also called "model choice" (267), which is confusing since they could refer to two different things (which model to use versus some decision made by a model), and the only way of knowing that this is actually a binary decision between an analytical or simulated annealing model is by reading the Note: in the user prompt appendix C.1 (L893-894).

Fragmented and missing details about the method, and inconsistent naming like this are common throughout the entire paper. I believe I understand a rough outline of the method, especially aided by reading Appendix C, but without being able to fully understand the method it is impossible to fairly evaluate the results and exactly what is being tested in the ablations.

**Questions:**

What exactly is encoded in the executed instances and circuit information of the standard database, and how is that encoded?

What is the exact definition of the Rank score? Is it a lexicographic ordering of Success Rate then HPWL?

Why is PeF the only direct comparison? It would be helpful to move some of the discussion of existing methods from Floorplan Backbones into the related work or background on EDA in order to help the reader understand the domain better. Introducing legalization earlier, and specifying that the no-overlap constraint is relaxed in this work's formulation would make references to Design Requirements throughout the methods section make more sense.

---

### Author Response · Authors · 2024-11-19
**Global Response**

Dear Area Chairs and Reviewers,

We sincerely appreciate all reviewers for your time and constructive suggestions. While the overall score of our submission is not as high as we had hoped, we are encouraged to see the reviewers acknowledge the importance of the problem (`CNXo`,`RMpB`), the extensive experimental study (`gWF3`), the strong performance of our approach (`CNXo`,`RMpB`), the novelty of our work (`gWF3`), and its potential to be both interesting and valuable (`tZbE`).

The most significant concern raised by reviewers is the organization of the paper. We recognize the challenge of integrating the fields of EDA, prompting, and agents within the constraints of a 10-page main text. This limitation may have led to an over-reliance on the Appendix to convey key information, resulting in confusion and some misunderstanding.

Regarding the given scores, we conclude that it is virtually impossible to achieve acceptance. Therefore, we have decided to withdraw the paper. That said, we remain confident in the value of our work and are committed to reorganizing and refining the manuscript for a stronger future submission.

Sincerely,
Authors of submission 6038

---

### Note · Authors · 2024-11-19

I have read and agree with the venue's withdrawal policy on behalf of myself and my co-authors.